# Posterior and Computational Uncertainty in Gaussian Processes

Jonathan Wenger[1,2]        Geoff Pleiss[2]

Marvin Pförtner[1]        Philipp Hennig[1,3]        John P. Cunningham[2]

[1] University of Tübingen
[2] Columbia University
[3] Max Planck Institute for Intelligent Systems, Tübingen

## Abstract

Gaussian processes scale prohibitively with the size of the dataset. In response, many approximation methods have been developed, which inevitably introduce approximation error. This additional source of uncertainty, due to limited computation, is entirely ignored when using the approximate posterior. Therefore in practice, GP models are often as much about the approximation method as they are about the data. Here, we develop a new class of methods that provides consistent estimation of the combined uncertainty arising from *both* the finite number of data observed *and* the finite amount of computation expended. The most common GP approximations map to an instance in this class, such as methods based on the Cholesky factorization, conjugate gradients, and inducing points. For any method in this class, we prove (i) convergence of its posterior mean in the associated RKHS, (ii) decomposability of its combined posterior covariance into mathematical and computational covariances, and (iii) that the combined variance is a tight worst-case bound for the squared error between the method's posterior mean and the latent function. Finally, we empirically demonstrate the consequences of ignoring computational uncertainty and show how implicitly modeling it improves generalization performance on benchmark datasets.

## 1   Introduction

Gaussian processes (GPs) are an expressive probabilistic model class, but their prohibitive scaling necessitates approximation [1]. A range of approximations based on kernel [2–10] or precision matrix [11–14] estimates, inducing point methods [15–22], and iterative solvers [23–29] have been proposed. These methods all use an affordable amount of computation to obtain an approximation of the *mathematical* posterior, which exists theoretically but cannot be accessed given limited computational resources. The approximate posterior is then used as a direct replacement of the mathematical posterior in downstream applications. Doing so, however, completely ignores the fact that we only expended a limited amount of compute. By analogy to the typical GP operation, where *limited data* induces modeling error captured by *mathematical uncertainty*, our work is motivated by the fact that *limited computation* induces approximation error that must be captured by *computational uncertainty*.

Here, we introduce IterGP, a class of methods which return a *combined uncertainty* that is the sum of mathematical and computational uncertainty. Figure 1 illustrates the difference between ignoring computational uncertainty and explicitly modeling it. We perform GP regression using a Matérn($\frac{3}{2}$) kernel on a toy dataset and compare SVGP (🔴) [22] to its analog in our framework, IterGP-PI (🟢 + 🔵), for a fixed set of inducing points. The computational shortcuts of inducing point methods can lead to unavoidable biases in their posterior mean and covariance [30, 31]. As Figure 1 illustrates, SVGP

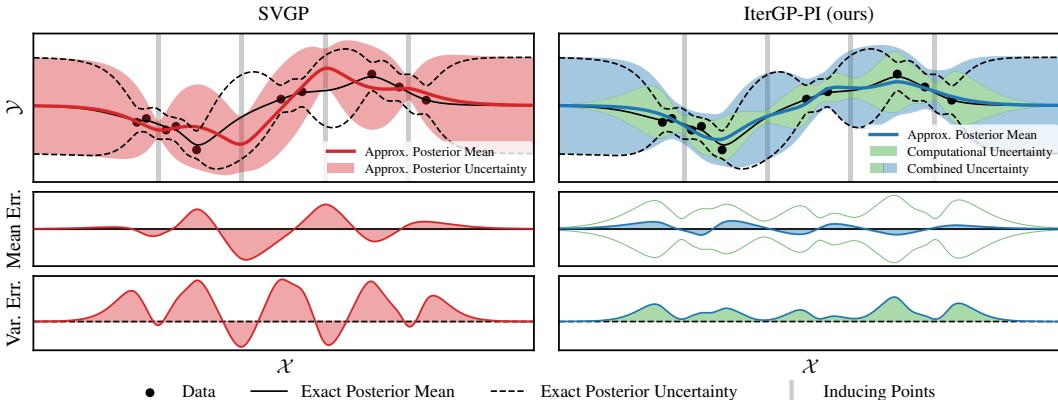

Figure 1: *Modeling computational uncertainty improves GP approximation.*

may underestimate the marginal variance where inducing points do not coincide with datapoints. In contrast, IterGP is guaranteed to overestimate the mathematical uncertainty – with the difference precisely given by the computational uncertainty (⬤). Additionally, the computational uncertainty is a worst-case bound (—) on the error of the approximate posterior mean.

To be clear, this overestimation is desirable: IterGP is not a typical approximation in the sense that its combined posterior attempts to approximate the mathematical posterior. Rather, IterGP recognizes that the mathematical posterior exists, but we do not have access to it, given computational constraints. Finite compute is as true a source of posterior uncertainty as finite data. Taking this view seriously, the true goal of GPs in the limited compute regime should in fact be to track combined uncertainty. This intuition motivates IterGP and is formally a feature of our results. We show that, if you update your GP via computation, specifically matrix-vector multiplication, then the combined uncertainty of the IterGP algorithm is precisely the correct object to capture your belief (Theorem 2) – in the same way the mathematical posterior is the correct object given finite data and unlimited computation.

Formally, IterGP is a probabilistic numerical method [32–35]. It treats the (unknown) representer weights as a latent variable with a prior belief that, when marginalized out, corresponds to a GP prior conditioned on no data. We then use a computational primitive (matrix-vector multiplication) that corresponds to tractable Bayesian updates on the representer weight distribution; that is, conditioning on computations performed on the data. The resulting belief can then be marginalized out to obtain a closed form, tractable expression for the combined – mathematical plus computational – uncertainty. This uncertainty quantification can be done *exactly* in quadratic time and linear space complexity.

Our framework admits three key theoretical properties. First, common GP approximations such as the partial Cholesky, the method of conjugate gradients and inducing point methods (e.g. SVGP) map to a corresponding IterGP instance. Therefore, these approaches can either be directly extended or modified to properly account for computational uncertainty. Second, the approximate posterior mean of any method in our proposed class converges to the mathematical posterior mean in RKHS norm in at most $n$ steps, where the convergence rate is determined by the choice of method (Theorem 1). Third, the combined uncertainty is a tight worst case bound on the relative error between the approximate posterior mean and the latent function (Theorem 2). To the best of our knowledge no existing GP approximation has this last property; an analogous guarantee only holds for exact GPs [36, Sec. 3.4].

**Contribution**  This work introduces IterGP, which defines a new class of GP approximations that accounts for computational uncertainty arising from limited computation. Some IterGP instances extend classic methods with improved uncertainty quantification (Table 1). For any method in this class, we prove that the approximate posterior mean converges to the mathematical posterior mean (Theorem 1) and that the combined uncertainty is a tight worst-case bound on the relative distance to the latent function one is trying to learn (Theorem 2, Corollary 1). We demonstrate empirically that modeling computational uncertainty can either save computation or improve generalization on a set of regression benchmark datasets. In conclusion, we show that it is possible to exactly quantify the inevitable error of GP approximations at quadratic cost by propagating said error to the posterior in the form of computational uncertainty.

## 2 Computation-Aware Gaussian Process Inference

We aim to learn a latent function $h : \mathcal{X} \to \mathcal{Y}$ from $\mathcal{X} \subseteq \mathbb{R}^d$ to $\mathcal{Y} \subseteq \mathbb{R}$ given a training dataset $\boldsymbol{X} = (\boldsymbol{x}_1, \dots, \boldsymbol{x}_n) \in \mathbb{R}^{n \times d}$ of $n$ inputs $\boldsymbol{x}_j \in \mathbb{R}^d$ and corresponding outputs $\boldsymbol{y} = (y_1, \dots, y_n)^\mathsf{T} \in \mathbb{R}^n$.

**Gaussian Processes** A stochastic process $f \sim \mathcal{GP}(\mu, k)$ with mean function $\mu : \mathbb{R}^d \to \mathbb{R}$ and kernel $k : \mathbb{R}^d \times \mathbb{R}^d \to \mathbb{R}$ is called a *Gaussian process* (GP) if any collection of function values $\mathbf{f} = (f(\boldsymbol{x}_1), \dots, f(\boldsymbol{x}_n))^\mathsf{T} \sim \mathcal{N}(\boldsymbol{\mu}, \boldsymbol{K})$ is jointly Gaussian with $\boldsymbol{\mu}_j = \mu(\boldsymbol{x}_j)$ and $\boldsymbol{K}_{ij} = k(\boldsymbol{x}_i, \boldsymbol{x}_j)$. Assuming observation noise $\boldsymbol{y} \mid \mathbf{f} \sim \mathcal{N}(\mathbf{f}, \sigma^2 \boldsymbol{I})$, the posterior distribution at test inputs $\boldsymbol{X}_\diamond$ is given by $\mathbf{f}_\diamond \sim \mathcal{N}(\mu_*(\boldsymbol{X}_\diamond), k_*(\boldsymbol{X}_\diamond, \boldsymbol{X}_\diamond))$ where the posterior mean and covariance functions are given by

$$\mu_*(\cdot) = \mu(\cdot) + k(\cdot, \boldsymbol{X}) \overbrace{\hat{\boldsymbol{K}}^{-1}(\boldsymbol{y} - \boldsymbol{\mu})}^{\boldsymbol{v}_*}, \quad \text{and} \quad k_*(\cdot, \cdot) = k(\cdot, \cdot) - k(\cdot, \boldsymbol{X}) \hat{\boldsymbol{K}}^{-1} k(\boldsymbol{X}, \cdot) \tag{1}$$

where $\hat{\boldsymbol{K}} := \boldsymbol{K} + \sigma^2 \boldsymbol{I} \in \mathbb{R}^{n \times n}$. Computing the *representer weights* $\boldsymbol{v}_* = \hat{\boldsymbol{K}}^{-1}(\boldsymbol{y} - \boldsymbol{\mu})$ exactly (as well as the posterior variance) is prohibitive given our limited computational budget.

**Learning Representer Weights** Consider the conditional distribution of the latent GP given its representer weights:

$$p(\mathbf{f}_\diamond \mid \boldsymbol{v}_*) = \mathcal{N}(\mu(\boldsymbol{X}_\diamond) + k(\boldsymbol{X}_\diamond, \boldsymbol{X})\boldsymbol{v}_*, \ k_*(\boldsymbol{X}_\diamond, \boldsymbol{X}_\diamond)). \tag{2}$$

When $\boldsymbol{v}_*$ is known exactly, we recover eq. (1). However, if we instead treat $\boldsymbol{v}_*$ as a random variable with the prior $p(\boldsymbol{v}_*) = \mathcal{N}(\boldsymbol{v}_*; \boldsymbol{0}, \hat{\boldsymbol{K}}^{-1})$, then the resulting marginal $\int p(\mathbf{f}_\diamond \mid \boldsymbol{v}_*) p(\boldsymbol{v}_*) \, d\boldsymbol{v}_*$ recovers the GP prior $\mathcal{N}(\mu(\boldsymbol{X}_\diamond), k(\boldsymbol{X}_\diamond, \boldsymbol{X}_\diamond))$. Our goal is to update this prior by iteratively applying the tractable computational primitive (i.e. matrix-vector multiplies). More specifically, each iteration conditions the current belief distribution $p(\boldsymbol{v}_*) = \mathcal{N}(\boldsymbol{v}_*; \boldsymbol{v}_{i-1}, \boldsymbol{\Sigma}_{i-1})$ on a one-dimensional projection of the current *residual* $\boldsymbol{r}_{i-1} = (\boldsymbol{y} - \boldsymbol{\mu}) - \hat{\boldsymbol{K}} \boldsymbol{v}_{i-1}$, where the projection is defined by an arbitrary vector $\boldsymbol{s}_i$:

$$\alpha_i := \boldsymbol{s}_i^\mathsf{T} \boldsymbol{r}_{i-1} = \boldsymbol{s}_i^\mathsf{T}((\boldsymbol{y} - \boldsymbol{\mu}) - \hat{\boldsymbol{K}} \boldsymbol{v}_{i-1}) = \boldsymbol{s}_i^\mathsf{T} \hat{\boldsymbol{K}}(\boldsymbol{v}_* - \boldsymbol{v}_{i-1}). \tag{3}$$

The choice of *actions* $\boldsymbol{s}_i$, which intuitively weight the approximation error of selected datapoints, defines different instances of our IterGP framework. Computing eq. (3) requires a single matrix-vector multiplication. After computing $\alpha_i$, we can perform an exact Bayesian update of $p(\boldsymbol{v}_*)$ via linear Gaussian identities. The updated $p(\boldsymbol{v}_*)$ (conditioned on $\alpha_i$) is $\mathcal{N}(\boldsymbol{v}_* \mid \boldsymbol{v}_i, \boldsymbol{\Sigma}_i)$, with

$$\boldsymbol{v}_i = \boldsymbol{v}_{i-1} + \underbrace{\boldsymbol{\Sigma}_{i-1} \hat{\boldsymbol{K}} \boldsymbol{s}_i}_{=: \boldsymbol{d}_i} \underbrace{(\boldsymbol{s}_i^\mathsf{T} \hat{\boldsymbol{K}} \boldsymbol{\Sigma}_{i-1} \hat{\boldsymbol{K}} \boldsymbol{s}_i)^{-1}}_{=: \eta_i} \underbrace{\boldsymbol{s}_i^\mathsf{T} \hat{\boldsymbol{K}}(\boldsymbol{v}_* - \boldsymbol{v}_{i-1})}_{= \alpha_i} = \boldsymbol{C}_i(\boldsymbol{y} - \boldsymbol{\mu}) \tag{4}$$

$$\boldsymbol{\Sigma}_i = \boldsymbol{\Sigma}_{i-1} - \underbrace{\boldsymbol{\Sigma}_{i-1} \hat{\boldsymbol{K}} \boldsymbol{s}_i}_{= \boldsymbol{d}_i} \underbrace{(\boldsymbol{s}_i^\mathsf{T} \hat{\boldsymbol{K}} \boldsymbol{\Sigma}_{i-1} \hat{\boldsymbol{K}} \boldsymbol{s}_i)^{-1}}_{= \eta_i} \underbrace{\boldsymbol{s}_i^\mathsf{T} \hat{\boldsymbol{K}} \boldsymbol{\Sigma}_{i-1}}_{= \boldsymbol{d}_i^\mathsf{T}} = \hat{\boldsymbol{K}}^{-1} - \boldsymbol{C}_i. \tag{5}$$

where $\boldsymbol{C}_i := \sum_{j=1}^i \frac{1}{\eta_j} \boldsymbol{d}_j \boldsymbol{d}_j^\mathsf{T} = \boldsymbol{S}_i(\boldsymbol{S}_i^\mathsf{T} \hat{\boldsymbol{K}} \boldsymbol{S}_i)^{-1} \boldsymbol{S}_i^\mathsf{T}$ is a rank-$i$ matrix (see Proposition S3 for details). With each computation, the uncertainty about the representer weights contracts as $\boldsymbol{C}_i \to \hat{\boldsymbol{K}}^{-1} = \boldsymbol{\Sigma}_0$ as $i \to n$. After $n$ iterations, $\boldsymbol{C}_n = \hat{\boldsymbol{K}}^{-1}$, meaning we fully recovered the representer weights with zero uncertainty. The consistent estimate for the representer weights is consequently $\boldsymbol{v}_i = \boldsymbol{C}_i(\boldsymbol{y} - \boldsymbol{\mu})$.

**Combining Mathematical and Computational Uncertainty** We now have a belief $p(\boldsymbol{v}_*) = \mathcal{N}(\boldsymbol{v}_*; \boldsymbol{v}_i, \boldsymbol{\Sigma}_i)$ about the representer weights reflecting the expended computation. To account for this computational uncertainty, we treat the representer weights as a latent variable of the mathematical posterior by reparameterizing $p(\mathbf{f}_\diamond \mid \boldsymbol{y}) = p(\mathbf{f}_\diamond \mid \boldsymbol{v}_*)$ and then marginalizing. The resulting marginal considers all possible representer weights which would have resulted in the same computational observations and therefore *implicitly* adds the uncertainty coming from *the computation itself*. Since the posterior mean of a GP is a linear function of the representer weights, the marginal distribution is given by $p(\mathbf{f}_\diamond) = \int p(\mathbf{f}_\diamond \mid \boldsymbol{v}_*) p(\boldsymbol{v}_*) \, d\boldsymbol{v}_* = \mathcal{N}(\mathbf{f}_\diamond; \mu_i(\boldsymbol{X}_\diamond), k_i(\boldsymbol{X}_\diamond, \boldsymbol{X}_\diamond))$, where

$$\mu_i(\cdot) = \mu(\cdot) + k(\cdot, \boldsymbol{X})\boldsymbol{v}_i$$

$$k_i(\cdot, \cdot) = \underbrace{k(\cdot, \cdot) - k(\cdot, \boldsymbol{X}) \hat{\boldsymbol{K}}^{-1} k(\boldsymbol{X}, \cdot)}_{\text{mathematical uncertainty} \ \color{blue}{\bullet}} + \underbrace{k(\cdot, \boldsymbol{X}) \boldsymbol{\Sigma}_i k(\boldsymbol{X}, \cdot)}_{\text{computational uncertainty} \ \color{green}{\bullet}} = \underbrace{k(\cdot, \cdot) - k(\cdot, \boldsymbol{X}) \boldsymbol{C}_i k(\boldsymbol{X}, \cdot)}_{\text{combined uncertainty} \ \color{purple}{\bullet}} \tag{6}$$

since $\boldsymbol{\Sigma}_i = \hat{\boldsymbol{K}}^{-1} - \boldsymbol{C}_i$.[1] As we perform more computation, the computational uncertainty reduces and we approach the mathematical uncertainty. We note that, while the *individual* terms are computationally prohibitive, the *combined* uncertainty can be evaluated cheaply since the approximate

---

[1] While we derive the combined posterior from a probabilistic numerics perspective, we can alternatively interpret eq. (6) as the GP prior $f$ conditioned on linearly transformed data $\boldsymbol{S}_i^\mathsf{T} \boldsymbol{y} \mid \mathbf{f} \sim \mathcal{N}(\boldsymbol{S}_i^\mathsf{T} \mathbf{f}, \sigma^2 \boldsymbol{S}_i^\mathsf{T} \boldsymbol{S}_i)$.

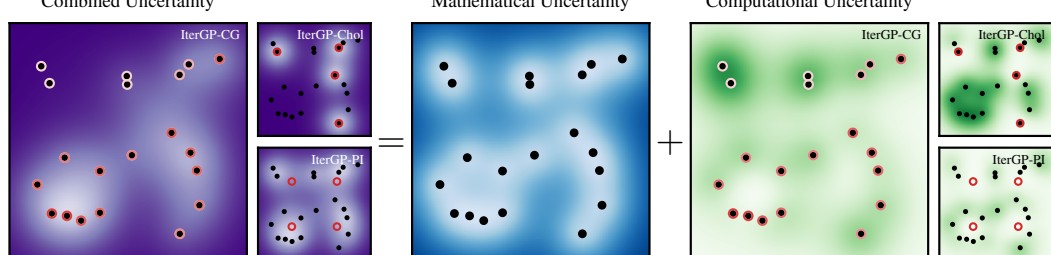

Figure 2: *Decomposition of the combined uncertainty.* The combined uncertainty (🟣) output by IterGP decomposes into the mathematical uncertainty (🔵) and computational uncertainty (🟢). After $i = 4$ iterations of Algorithm 1 computational uncertainty is small in parts of the input space where there either is no data (●) or computation was "targeted" (⊙). Which datapoints are targeted in each iteration and to what degree is defined by the magnitude of the action vector elements $(s_i)_j$. Different instances of IterGP either reduce computational uncertainty locally (e.g. IterGP-Chol, IterGP-PI) or globally (e.g. IterGP-CG). After $n$ iterations the mathematical uncertainty is recovered.

precision matrix $C_i$ is of low rank. Figure 2 illustrates that computational uncertainty is large where there are data and we have not targeted computation yet. Different methods from our proposed class target computation in different parts of the input space. Where there is no data the prior is a good approximation of the posterior and therefore computational uncertainty is low.

Algorithm 1 computes an estimate of the representer weights $v_i$ and the rank-$i$ precision matrix approximation $C_i$. A specific instance of IterGP is defined by a sequence of actions $s_i$. To gain an intuition for how Algorithm 1 operates, it helps to interpret it as targeting a given computational budget towards certain datapoints defined by $s_i$. Near datapoints $x_j$ that are not targeted, i.e. $(s_i)_j = 0$, computational uncertainty remains unchanged. In fact, datapoints $(x_j, y_j)$ that are never targeted up to iteration $i$ are not needed to compute $\mathcal{GP}(\mu_i, k_i)$, meaning that Algorithm 1 is *inherently online* and we can *observe data sequentially* without having to restart the algorithm (see Theorem S7).

---

**Algorithm 1: A Class of Computation-Aware Iterative Methods for GP Approximation**

---

**Input:** prior mean function $\mu$, prior covariance function / kernel $k$, training inputs $X$, labels $y$
**Output:** (combined) GP posterior $\mathcal{GP}(\mu_i, k_i)$

1   **procedure** IterGP($\mu, k, X, y$)
2     $(\mu_0, k_0) \leftarrow (\mu, k)$                          ▷ Initialize mean and covariance function with prior.
3     $\boldsymbol{\mu} \leftarrow \mu(X)$                                  ▷ Prior predictive mean.
4     $\hat{K} \leftarrow k(X, X) + \sigma^2 I$                      ▷ Prior predictive kernel matrix.
5     **while not** StoppingCriterion() **do**                ▷ Stopping criterion.
6       $s_i \leftarrow$ Policy()             ▷ Select action via policy (see Table 1 for examples).
7       $r_{i-1} \leftarrow (y - \boldsymbol{\mu}) - \hat{K} v_{i-1}$            ▷ Predictive residual.
8       $\alpha_i \leftarrow s_i^\mathsf{T} r_{i-1}$             ▷ Observation via information operator.
9       $d_i \leftarrow \Sigma_{i-1} \hat{K} s_i = (I - C_{i-1} \hat{K}) s_i$       ▷ Search direction.
10      $\eta_i \leftarrow s_i^\mathsf{T} \hat{K} \Sigma_{i-1} \hat{K} s_i = s_i^\mathsf{T} \hat{K} d_i$      ▷ Normalization constant.
11      $C_i \leftarrow C_{i-1} + \frac{1}{\eta_i} d_i d_i^\mathsf{T}$       ▷ Precision matrix approximation $C_i \approx \hat{K}^{-1}$.
12      $Q_i \leftarrow Q_{i-1} + \frac{1}{\eta_i} \hat{K} d_i d_i^\mathsf{T} \hat{K}$       ▷ Kernel matrix approximation $Q_i \approx \hat{K}$.
13      $v_i \leftarrow v_{i-1} + \frac{\alpha_i}{\eta_i} d_i$          ▷ Representer weights estimate.
14      $\Sigma_i \leftarrow \Sigma_0 - C_i$          ▷ Computational representer weights uncertainty.
15      $p(v_*) \leftarrow \mathcal{N}(v_*; v_i, \Sigma_i)$          ▷ Belief about representer weights.
16      $\mu_i(\cdot) \leftarrow \mu(\cdot) + k(\cdot, X) v_i$         ▷ Approximate posterior mean function.
17      $k_i(\cdot, \cdot) \leftarrow k(\cdot, \cdot) - k(\cdot, X) C_i k(X, \cdot)$       ▷ Combined uncertainty.
18     **return** $\mathcal{GP}(\mu_i, k_i)$

---

Greyed out quantities are *not* needed to compute the combined posterior and are only included for clarity of exposition.

Table 1: *Instances of Algorithm 1, which map to commonly used GP approximations.*

| Method | Actions $\boldsymbol{s}_i$ | Classic Analog | Reference |
|---|---|---|---|
| IterGP-Chol | $\boldsymbol{e}_i$ | (partial) Cholesky | Theorem S3 |
| IterGP-PBR | $\text{ev}_i(\hat{\boldsymbol{K}})$ | (partial) EVD / SVD | Theorem S4 |
| IterGP-CG | $\boldsymbol{s}_i^{\text{PCG}}$ or $\hat{\boldsymbol{P}}^{-1}\boldsymbol{r}_i$ | (preconditioned) CG | Theorem S5 and Corollary S2 |
| IterGP-PI | $k(\boldsymbol{X}, \boldsymbol{z}_i)$ | $\approx$ Nyström (SoR, DTC), SVGP | Section 2.1 and Theorem S6 |

## 2.1 Connection to Other GP Approximation Methods

IterGP extends the most commonly used GP approximations to include computational uncertainty, with at most quadratic cost (see Table 1 for a summary and Figure 2, Figure S3 for illustration).

**Cholesky Decomposition** The (partial) Cholesky decomposition iteratively chooses datapoints or pivots $\boldsymbol{x}_i$ based on a given ordering. The resulting Cholesky factor is lower triangular and increases in rank each iteration, and a well-chosen ordering achieves fast convergence (cf. [37, Thm. 2.3]). If one chooses standard unit vectors $\boldsymbol{e}_i$ as actions corresponding to the selected datapoint per iteration, then Algorithm 1 recovers the partial Cholesky factorization exactly (Theorem S3).

**Conjugate Gradients** CG [38] with preconditioning for GP inference has become increasingly popular [24–29, 39, 40]. Algorithm 1 recovers preconditioned CG exactly, if we choose either preconditioned conjugate gradients or residuals as actions (see Theorem S5 and Corollary S2). In fact, Algorithm 1 can even construct its own diagonal-plus-low-rank preconditioner by first running a few iterations with an arbitrary policy and then using the byproducts of these iterations for the preconditioner. For example, if we run IterGP-Chol initially, we can construct an incomplete Cholesky preconditioner for subsequent CG iterations.

**Inducing Point Methods** Inducing point methods, such as variants of the Nyström approximation [16], i.e. subset of regressors (SoR) [15, 41] and deterministic training conditional (DTC) [18, 42], as well as SVGP [22] share a posterior mean, which by Theorem S6 takes the form

$$\mu_{\text{SVGP}}(\cdot) = q(\cdot, \boldsymbol{X})\boldsymbol{K}_{\boldsymbol{X}\boldsymbol{Z}}(\boldsymbol{K}_{\boldsymbol{Z}\boldsymbol{X}}(q(\boldsymbol{X}, \boldsymbol{X}) + \sigma^2\boldsymbol{I})\boldsymbol{K}_{\boldsymbol{X}\boldsymbol{Z}})^{-1}\boldsymbol{K}_{\boldsymbol{Z}\boldsymbol{X}}(\boldsymbol{y} - \boldsymbol{\mu}) \tag{7}$$

where $\boldsymbol{Z} \in \mathbb{R}^{n \times i}$ is a set of inducing points and $q(\cdot, \cdot) = k(\cdot, \boldsymbol{Z})\boldsymbol{K}_{\boldsymbol{Z}\boldsymbol{Z}}^{-1}k(\boldsymbol{Z}, \cdot)$. These approximations also have very closely related posterior covariance functions [20, 43]. If we choose actions $\boldsymbol{s}_i = k(\boldsymbol{X}, \boldsymbol{z}_i)$, by Proposition S3, Algorithm 1 returns a posterior mean given by

$$\mu_i(\cdot) = k(\cdot, \boldsymbol{X})\boldsymbol{K}_{\boldsymbol{X}\boldsymbol{Z}}\underbrace{(\boldsymbol{K}_{\boldsymbol{Z}\boldsymbol{X}}(k(\boldsymbol{X}, \boldsymbol{X}) + \sigma^2\boldsymbol{I})\boldsymbol{K}_{\boldsymbol{X}\boldsymbol{Z}})^{-1}}_{\text{Gram matrix } \boldsymbol{S}_i^{\mathsf{T}}\hat{\boldsymbol{K}}\boldsymbol{\Sigma}_0\hat{\boldsymbol{K}}\boldsymbol{S}_i}\boldsymbol{K}_{\boldsymbol{Z}\boldsymbol{X}}(\boldsymbol{y} - \boldsymbol{\mu}). \tag{8}$$

Choosing such actions, given by kernel functions $k(\cdot, \boldsymbol{z}_i)$ centered at inducing points $\boldsymbol{z}_i$, reduces computational uncertainty in regions close to inducing points (see IterGP-PI in Figure 2), where closeness is determined by the kernel. Comparing SVGP's and IterGP-PI's posterior mean provides a probabilistic numerical perspective on why even for small KL-divergence between the approximating distribution of SVGP and the true posterior, the mean estimate can be far from the true mean [31, Prop. 3.1]. As outlined in Section 2, eq. (8) is a Bayesian update on the initially unknown representer weights $\boldsymbol{v}_* = \hat{\boldsymbol{K}}^{-1}(\boldsymbol{y} - \boldsymbol{\mu})$. The Gram matrix in eq. (8) describes how surprising the computational observations $\boldsymbol{K}_{\boldsymbol{Z}\boldsymbol{X}}(\boldsymbol{y} - \boldsymbol{\mu}) = \boldsymbol{S}_i^{\mathsf{T}}(\boldsymbol{y} - \boldsymbol{\mu}) = \boldsymbol{S}_i^{\mathsf{T}}\hat{\boldsymbol{K}}\boldsymbol{v}_*$ of the representer weights should be, given the prior uncertainty $\boldsymbol{\Sigma}_0$ about them. SVGP uses a similar form for the posterior mean (c.f. (7) and (8)), but the Gram matrix is "smaller" since $q(\boldsymbol{X}, \boldsymbol{X}) \preceq k(\boldsymbol{X}, \boldsymbol{X})$. This can be interpreted as inducing point methods being overconfident in their update of the representer weight estimates to achieve linear time complexity. As the inducing points approach the data points the two posterior mean functions $\mu_{\text{SVGP}}$ and $\mu_i$ become closer and are equivalent if the inducing points equal the training data.

## 2.2 The Cost of Computational Uncertainty

Quantifying combined uncertainty has greater cost than linear time GP approximations such as inducing point methods, due to its use of matrix-vector multiplication as the computational operation to condition on the data. Algorithm 1 in its most general form performs three matrix-vector products per iteration resulting in a quadratic time complexity $\mathcal{O}(n^2 i)$ overall for $i$ iterations. In this sense,

Algorithm 1 represents a middle ground between the mathematical posterior—which incurs a cubic time complexity—and $\mathcal{O}(ni^2)$ approximations—which can only estimate their computational error through potentially loose theoretical bounds which may [e.g. 21, 22, 44] or may not be computable in less than $\mathcal{O}(n^3)$ [4, 37]. At any point during a run of Algorithm 1, computing the predictive mean on $n_\diamond$ new data points has cost $\mathcal{O}(n_\diamond n)$, while the marginal predictive (co-)variance can be evaluated in $\mathcal{O}(n_\diamond ni)$ since $\boldsymbol{C}_i$ is of rank $i$. Additionally, using Matheron's rule [45–47], sampling from the approximate posterior at $n_\diamond$ evaluation points also only requires $\mathcal{O}(n_\diamond ni)$ computation (assuming we can sample from the prior—see Section S3.3). The objects required to make predictions and draw samples are the vector $\boldsymbol{v}_i$ and low rank matrix $\boldsymbol{C}_i$ which both require $\mathcal{O}(ni)$ memory. Finally, the memory cost of Algorithm 1 is only linear in $n$, since matrix multiplication $\boldsymbol{v} \mapsto \hat{\boldsymbol{K}}\boldsymbol{v}$ can be computed without explicitly forming $\hat{\boldsymbol{K}}$ [48].

## 2.3 Related Work

GP inference based on matrix-vector multiplies, particularly CG [38], has become popular recently [5, 24–29, 39]. Advances in specialized hardware has boosted their scalability without excessive memory footprint [27, 48]. Such iterative methods typically rely on preconditioning, which has been shown to significantly improve their performance [25, 26, 29]. Our method generalizes CG in this setting and thus retains the same benefits. At its core Algorithm 1 employs a (Bayesian) probabilistic numerical method [32–35], more specifically a probabilistic linear solver (PLS) [49–54] applied to the linear system $\hat{\boldsymbol{K}}\boldsymbol{v}_* = \boldsymbol{y}$. The fact that a PLS using CG actions can recover CG in its posterior mean was observed previously [49, 51, 53]. Here, we extend this result to residual actions and preconditioning. Further, we also demonstrate the connection to the Cholesky and singular value decompositions. For randomized actions, the PLS as part of Algorithm 1 also recovers the randomized Kaczmarz method in its posterior mean [55–58]. Employing a PLS for GP approximation by updating beliefs over the kernel and precision matrix was suggested previously [53, 59]. Our work differs in that it updates a belief over the representer weights, as opposed to the kernel function or matrix, considers more general projections than just conjugate residuals, and, most importantly, provides a theoretically motivated combined posterior which can be computed exactly.

# 3 Theoretical Analysis

The main goals of our theoretical analysis will be to prove

    (a) *convergence of IterGP's posterior mean* in norm (Theorem 1) and pointwise (Corollary 1)

and to provide rigorous justification for the combined and computational uncertainty. Importantly, the

    (b) *combined uncertainty is a tight worst-case bound on the relative distance to all potential latent functions* consistent with our (computational) observations (Theorem 2).

We will demonstrate a similar interpretation of the computational uncertainty as a bound on the relative error to the mathematical posterior mean (see eqs. (14) and (16)).

## 3.1 Estimation of Representer Weights

At the heart of Algorithm 1 is a probabilistic linear solver [49–51, 53] iteratively updating a belief about the representer weights. It constructs an expanding subspace $\text{span}\{\boldsymbol{s}_1, \ldots, \boldsymbol{s}_i\} = \text{span}\{\boldsymbol{d}_1, \ldots, \boldsymbol{d}_i\}$ spanned by the actions in which the inverse $\hat{\boldsymbol{K}}^{-1}$ is perfectly identified. Each step $\boldsymbol{d}_i$ expanding this explored subspace is $\hat{\boldsymbol{K}}$-orthogonal to the previous ones.

**Proposition 1** (Conjugate Direction Method)
*Let the actions $\boldsymbol{s}_i$ of Algorithm 1 be linearly independent. Then Algorithm 1 is a conjugate direction method, i.e. it holds that $\boldsymbol{d}_i^\mathsf{T} \hat{\boldsymbol{K}} \boldsymbol{d}_j = 0$ for all $i \neq j$.*

*Proof.* Without loss of generality assume $i > j$. Then the result follows directly from Lemma S1. $\square$

Geometrically, Algorithm 1 iteratively projects the representer weights onto the expanding subspace $\text{span}\{\boldsymbol{S}_i\}$ with respect to $\langle \cdot, \cdot \rangle_{\hat{K}}$. We can use this intuition to understand the convergence of the representer weights estimate. The relative error $\rho(i)$ at iteration $i$ is given by how small the "angle" between this subspace and the representer weights vector is.

**Proposition 2** (Relative Error Bound for the Representer Weights)
*For any choice of actions a relative error bound $\rho(i)$, s.t. $\|\boldsymbol{v}_* - \boldsymbol{v}_i\|_{\hat{\boldsymbol{K}}} \leq \rho(i)\|\boldsymbol{v}_*\|_{\hat{\boldsymbol{K}}}$ is given by*

$$\rho(i) = (\bar{\boldsymbol{v}}_*^\mathsf{T} \underbrace{(\boldsymbol{I} - \boldsymbol{C}_i\hat{\boldsymbol{K}})}_{\text{projection onto } \text{span}\{\boldsymbol{S}_i\}^{\perp \hat{\boldsymbol{K}}}} \bar{\boldsymbol{v}}_*)^{\frac{1}{2}} \leq \lambda_{\max}(\boldsymbol{I} - \boldsymbol{C}_i\hat{\boldsymbol{K}}) \leq 1 \tag{9}$$

*where $\bar{\boldsymbol{v}}_* = \boldsymbol{v}_*/\|\boldsymbol{v}_*\|_{\hat{\boldsymbol{K}}}$. If the actions $\{\boldsymbol{s}_i\}_{i=1}^n$ are linearly independent, then $\rho(i) \leq \delta_{n=i}$.*

*Proof.* See Section S2.2. $\qquad\square$

Proposition 2 guarantees convergence in at most $n$ iterations, if the actions are chosen to be linearly independent, since $\boldsymbol{C}_i\hat{\boldsymbol{K}}$ is a $\hat{\boldsymbol{K}}$-orthogonal projection onto $\text{span}\{\boldsymbol{S}_i\}$ (see Lemma S1). Therefore, if our finite computational budget is large enough, we eventually recover the mathematical posterior. This is reflected by the contraction of the posterior over the representer weights (see Proposition S4). The bound in Proposition 2 is tight without further assumptions on the actions, since there exists an adverserial sequence of actions such that the first $(n-1)$ are in $\text{span}\{\boldsymbol{v}_*\}^{\perp \hat{\boldsymbol{K}}}$. Then the inverse is perfectly identified in that subspace, but $\boldsymbol{v}_i = \boldsymbol{C}_i\boldsymbol{y} = \boldsymbol{C}_i\hat{\boldsymbol{K}}\boldsymbol{v}_* = \boldsymbol{0}$. In practice, one can derive tighter convergence bounds for specific sequences of actions. For example, for randomized actions the bound depends on their distribution [56, 57]. If residuals $\boldsymbol{r}_i$ are chosen as actions, we obtain

$$\rho(i) = 2\left(\frac{\sqrt{\kappa}-1}{\sqrt{\kappa}+1}\right)^i \text{ or } \rho(i) = \left(\frac{\lambda_{n-i}-\lambda_1}{\lambda_{n-i}+\lambda_1}\right) \tag{10}$$

since then Algorithm 1's estimate of the representer weights equals that of CG (Corollary S2). Here $\kappa$ is the condition number and $\lambda_j$ the eigenvalues of either (i) the kernel matrix $\hat{\boldsymbol{K}}$ if $\boldsymbol{s}_i = \boldsymbol{r}_i$, or (ii) the preconditioned kernel matrix $\hat{\boldsymbol{P}}^{-\frac{1}{2}}\hat{\boldsymbol{K}}\hat{\boldsymbol{P}}^{-\frac{1}{2}}$ if $\boldsymbol{s}_i = \hat{\boldsymbol{P}}^{-1}\boldsymbol{r}_i$.

## 3.2 Convergence in RKHS Norm of the Posterior Mean

Having established convergence of the representer weights estimate, we can use this result to prove convergence in norm of IterGP's posterior mean to the mathematical posterior at the same rate.

**Theorem 1** (Convergence in RKHS Norm of the Posterior Mean Approximation)
*Let $\mathcal{H}_k$ be the RKHS associated with kernel $k(\cdot, \cdot)$, $\sigma^2 > 0$ and let $\mu_* - \mu \in \mathcal{H}_k$ be the unique solution to the regularized empirical risk minimization problem*

$$\arg\min_{f \in \mathcal{H}_k} \tfrac{1}{n}\left(\sum_{j=1}^n (f(\boldsymbol{x}_j) - y_j + \mu(\boldsymbol{x}_j))^2 + \sigma^2\|f\|_{\mathcal{H}_k}^2\right) \tag{11}$$

*which is equivalent to the mathematical posterior mean up to shift by the prior $\mu$ [e.g. 1, Sec. 6.2]. Then for $i \in \{0, \ldots, n\}$ the posterior mean $\mu_i(\cdot)$ computed by Algorithm 1 satisfies*

$$\boxed{\|\mu_* - \mu_i\|_{\mathcal{H}_k} \leq \rho(i)c(\sigma^2)\|\mu_* - \mu_0\|_{\mathcal{H}_k}} \tag{12}$$

*where $\mu_0 = \mu$ is the prior mean and the constant $c(\sigma^2) = \sqrt{1 + \frac{\sigma^2}{\lambda_{\min}(\boldsymbol{K})}} \to 1$ as $\sigma^2 \to 0$.*

*Proof.* See Section S2.3. $\qquad\square$

Theorem 1 gives a bound on the RKHS-norm error between the posterior mean $\mu_i$ of IterGP and the mathematical posterior mean $\mu_*$. If for the given prior kernel a bound on the RKHS-norm error $\|h - \mu_*\|_{\mathcal{H}_k}$ between the latent function $h$ and the mathematical posterior mean $\mu_*$ is known, Theorem 1 can be directly used to bound the RKHS-norm error between IterGP's posterior mean and the latent function $h$ via the triangle inequality: $\|h - \mu_i\|_{\mathcal{H}_k} \leq \underbrace{\|h - \mu_*\|_{\mathcal{H}_k}}_{\to 0 \text{ as } n \to \infty} + \underbrace{\|\mu_* - \mu_i\|_{\mathcal{H}_k}}_{\to 0 \text{ as } i \to n}$.

## 3.3 Combined and Computational Uncertainty as Worst Case Errors

While Theorem 1 shows convergence in norm for IterGP's posterior mean, the convergence rate $\rho(i)$ may contain expressions which cannot be evaluated at runtime with the limited computation at our disposal. For example, for residual actions evaluating eq. (10) requires computation of the kernel matrix spectrum. However, the combined uncertainty of IterGP is a tight bound on the *pointwise* relative error to all possible latent functions which would have resulted in the same computations.

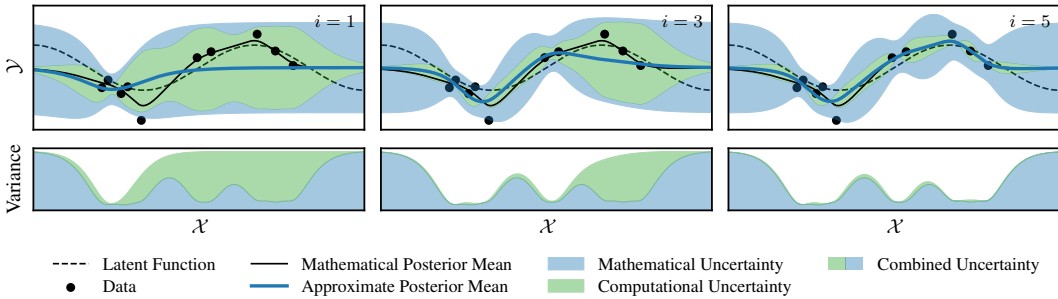

Figure 3: *Computational and combined uncertainty of IterGP as worst-case bounds.*[2]

**Theorem 2** (Combined and Computational Uncertainty as Worst Case Errors)
*Let $\sigma^2 \geq 0$ and let $k_i(\cdot, \cdot) = k_*(\cdot, \cdot) + k_i^{\text{comp}}(\cdot, \cdot)$ be the combined uncertainty computed by Algorithm 1. Then, for any $\boldsymbol{x} \in \mathcal{X}$ (assuming $\boldsymbol{x} \notin \boldsymbol{X}$ if $\sigma^2 > 0$) we have*

$$\sup_{g \in \mathcal{H}_{k\sigma}:\|g\|_{\mathcal{H}_{k\sigma}} \leq 1} \overbrace{\underbrace{g(\boldsymbol{x}) - \mu_*^g(\boldsymbol{x})}_{\text{error of math. post. mean } \bigcirc} + \underbrace{\mu_*^g(\boldsymbol{x}) - \mu_i^g(\boldsymbol{x})}_{\text{computational error } \bigcirc}}^{\text{error of approximate posterior mean } \bigcirc} = \sqrt{k_i(\boldsymbol{x}, \boldsymbol{x}) + \sigma^2}, \quad and \quad (13)$$

$$\sup_{g \in \mathcal{H}_{k\sigma}:\|g\|_{\mathcal{H}_{k\sigma}} \leq 1} \underbrace{\mu_*^g(\boldsymbol{x}) - \mu_i^g(\boldsymbol{x})}_{\text{computational error } \bigcirc} = \sqrt{k_i^{\text{comp}}(\boldsymbol{x}, \boldsymbol{x})} \quad (14)$$

*where $\mu_*^g(\cdot) = k(\cdot, \boldsymbol{X})\hat{\boldsymbol{K}}^{-1}g(\boldsymbol{X})$ is the mathematical and $\mu_i^g(\cdot) = k(\cdot, \boldsymbol{X})\boldsymbol{C}_i g(\boldsymbol{X})$ IterGP's posterior mean for the latent function $g \in \mathcal{H}_{k\sigma}$. If $\sigma^2 = 0$, then the above also holds for $\boldsymbol{x} \in \boldsymbol{X}$.*

*Proof.* See Section S2.4. □

Theorem 2 rigorously explains why the combined (mathematical + computational) uncertainty $k_i$ is the correct object characterizing our belief about the latent function $h$, given that we are in the limited compute regime. In the same way that the mathematical uncertainty is a tight bound on the distance to all functions $g$ which could have produced the data (see [36, Prop. 3.8]), the combined uncertainty is a tight bound on all functions $g$ which would have produced the same computations.

### 3.4 Pointwise Convergence of the Posterior Mean

In particular, as Corollary 1 shows and Figure 3 illustrates, the computational uncertainty ($\bigcirc$) is a pointwise bound on the relative distance to the mathematical posterior mean (16) and *the combined uncertainty ($\bigcirc$ + $\bigcirc$) is a pointwise bound on the relative distance to the true latent function* (15).

**Corollary 1** (Pointwise Convergence of the Posterior Mean)
*Assume the conditions of Theorem 2 hold and assume the latent function $h \in \mathcal{H}_{k\sigma}$. Let $\mu_*$ be the corresponding mathematical posterior mean and $\mu_i$ the posterior mean computed by Algorithm 1. Then it holds that*

$$\frac{|h(\boldsymbol{x}) - \mu_i(\boldsymbol{x})|}{\|h\|_{\mathcal{H}_{k\sigma}}} \leq \sqrt{k_i(\boldsymbol{x}, \boldsymbol{x}) + \sigma^2}, \quad and \quad (15)$$

$$\frac{|\mu_*(\boldsymbol{x}) - \mu_i(\boldsymbol{x})|}{\|h\|_{\mathcal{H}_{k\sigma}}} \leq \sqrt{k_i^{\text{comp}}(\boldsymbol{x}, \boldsymbol{x})}. \quad (16)$$

*Proof.* This follows immediately from Theorem 2 by recognizing that $h/\|h\|_{\mathcal{H}_{k\sigma}}$ has unit norm. □

---

[2]The combined (co-)variance decomposes into mathematical and computational covariances, as opposed to the combined standard deviation since $\sqrt{\bigcirc + \bigcirc} \neq \sqrt{\bigcirc} + \sqrt{\bigcirc}$. The bottom panel thus illustrates the variance decomposition. However, to better illustrate Theorem 2, in the upper panel we plot the combined standard deviation $\sqrt{\bigcirc + \bigcirc}$ and computational standard deviation $\sqrt{\bigcirc}$ within it, in line with standard GP plotting practice.

It is worth noting that Theorem 2 and Corollary 1 generally *do not hold for other GP approximations*. They explicitly rely on $C_i \hat{K}$ being the $\hat{K}$-orthogonal projection onto the space spanned by the actions (see Lemma S1). Since orthogonal projections are unique, if another GP approximation is such a projection and therefore satisfies Theorem 2, it is in fact an instance of IterGP.

## 4 Experiments

To demonstrate the effects of quantifying computational uncertainty we perform GP regression on synthetic and benchmark datasets for the two most common GP approximations in the large-scale setting, SVGP [22] and CGGP [26], and their direct analogs from our class of methods. An implementation of Algorithm 1, based on KeOps [48] and ProbNum [60], is available at:

https://github.com/JonathanWenger/itergp

**Experimental Setup** We consider a synthetic dataset of iid uniformly sampled inputs $\boldsymbol{x}_j \in [-1, 1]^d$ with $y(\boldsymbol{x}) = \sin(\pi \boldsymbol{x}^\intercal \mathbf{1}) + \varepsilon$, where $\varepsilon \sim \mathcal{N}(0, \sigma^2)$, as well as a range of UCI datasets [61] with training set sizes $n = 5,287$ to $57,247$, dimensions $d = 9$ to $26$ and standardized features. All experiments were run on an NVIDIA GeForce RTX 2080 Ti graphics card. We perform GP regression using a zero mean prior and a Matérn($\frac{1}{2}$) kernel (for other kernels see Section S4). All experiments were run 10 times with randomly sampled training and test splits of $90/10$ and we report average metrics with 95% confidence intervals.

**IterGP reduces the necessary computations for CG-based GP inference.** We compare IterGP to the CG-based GP inference used in the GPyTorch library [26]. For all datasets, we select hyperparameters using the training procedure of Wenger et al. [29]. As we show in Theorem S5, the posterior mean of IterGP with (conjugate) residual actions is exactly equivalent to performing CG to compute the representer weights. Therefore, both methods produce the exact same posterior mean estimate and thus achieve the same RMSE as a function of CG iterations (Figure 4, bottom). The primary difference between the two methods is in the posterior variance. The combined variance estimate of IterGP is essentially "free" in the sense that it reuses terms from the posterior mean calculation. In contrast, computing the posterior variance with CG requires $n_\diamond$ additional linear solves ($\hat{K}^{-1}\boldsymbol{x}_{\diamond 1}, \ldots, \hat{K}^{-1}\boldsymbol{x}_{\diamond n_\diamond}$). GPyTorch relies on the Lanczos Variance Estimate technique [62] which essentially warm-starts each of these solves by reusing quantities from the linear solve $\hat{K}^{-1}k(\boldsymbol{X}, \boldsymbol{x}_{\diamond 1})$. While this approach produces reliable variance estimates that converge to the true posterior variance, it requires additional computation: at least one set of additional CG iterations to compute $\hat{K}^{-1}k(\boldsymbol{X}, \boldsymbol{x}_{\diamond 1})$. In Figure 4(a) (top), we see that IterGP and GPyTorch's CGGP achieve nearly identical NLL, suggesting that both methods produce variances that yield similar generalization.

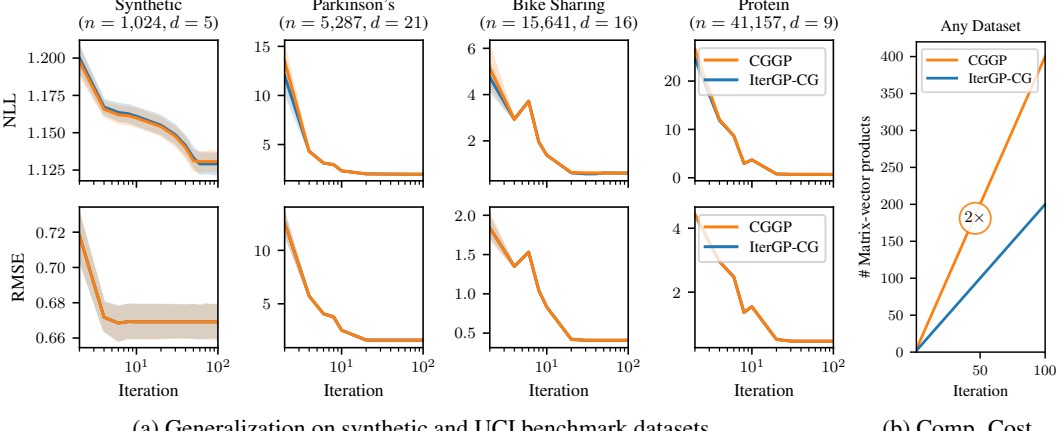

(a) Generalization on synthetic and UCI benchmark datasets.  (b) Comp. Cost

Figure 4: *Generalization of CGGP and its closest IterGP analog.* (a) GP regression using a Matérn($\frac{1}{2}$) kernel on UCI datasets. The plot shows the average generalization error in terms of NLL and RMSE for an increasing number of solver iterations. The posterior mean of IterGP-CG and CGGP is identical, which explains the identical RMSE. However, CGGP performs additional computation for the posterior covariance as (b) illustrates, which is not needed since IterGP-CG has identical NLL.

The key difference between the methods is that 1) unlike CGGP, IterGP's variances exactly capture both mathematical and computational uncertainty, and 2) IterGP's variances require no additional solves, resulting in half as much computation as GPyTorch's CGGP implementation (see Figure 4(b)).

**Quantifying computational uncertainty improves generalization of inducing point methods.** To understand the benefits of quantifying computational uncertainty, we compare the linear-time SVGP method (which does not quantify computational uncertainty) with the closest (quadratic-time) inducing point analog from our proposed IterGP framework (see Section 2.1). While the IterGP method is inherently more expensive than SVGP, our goal is simply to demonstrate that inducing points can yield far more accuracy if one has the budget to account for computational uncertainty. To that end, we compare SVGP against IterGP using the same set of randomly-placed inducing points. We identify a set of kernel hyperparameters by optimizing the ELBO of SVGP on the training data, using these for both SVGP and IterGP. As Figure 5 shows, we find that across all datasets that IterGP offers better RMSE and NLL than SVGP, despite the fact that the hyperparameters are chosen to favor SVGP. This suggests that the extra computation needed to quantify computational uncertainty can more "effectively" utilize a set of inducing points for predictive models.

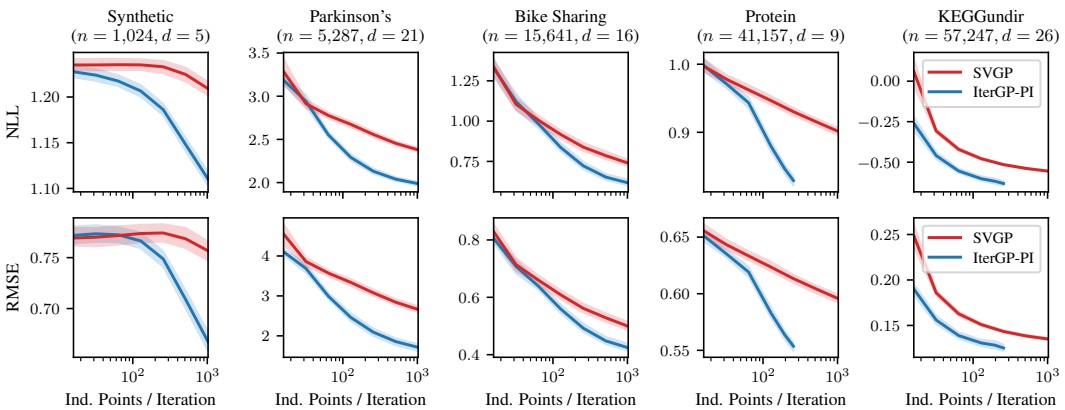

Figure 5: *Generalization of SVGP and its closest IterGP analog.* GP regression using a Matérn($\frac{1}{2}$) kernel on UCI datasets. The plot shows the average generalization error in terms of NLL and RMSE for an increasing number of identical inducing points. After a small number of inducing points relative to the size of the training data, IterGP has significantly lower generalization error than SVGP.

## 5  Conclusion

Scalable GP approximations inevitably introduce error, leading to a worse model for the latent function in question. This work demonstrates that it is possible to account for *both* uncertainty arising from limited data *and* uncertainty arising from limited computation *exactly* – which as we show improves model performance. IterGP methods return this combined uncertainty which crucially represents a dataset-specific, pointwise worst-case bound on the error to the true latent function. At its core, IterGP performs repeated matrix-vector multiplication resulting in quadratic complexity. Since modern computing architectures (i.e. GPUs) have been specifically designed for this operation at scale, iterative approaches for GP approximation are becoming competitive with theoretically cheaper approximations, like inducing point methods [26, 27]. Finally, in addition to the general utility of IterGP, we expect this class of methods to be particularly useful in applications where accurate uncertainty quantification is important or, due to its inherently online nature, where data is acquired sequentially such as in active learning and Bayesian optimization.

## Acknowledgments and Disclosure of Funding

JW, MP and PH gratefully acknowledge financial support by the European Research Council through ERC StG Action 757275 / PANAMA; the DFG Cluster of Excellence "Machine Learning - New Perspectives for Science", EXC 2064/1, project number 390727645; the German Federal Ministry of Education and Research (BMBF) through the Tübingen AI Center (FKZ: 01IS18039A); and funds from the Ministry of Science, Research and Arts of the State of Baden-Württemberg. The authors thank the International Max Planck Research School for Intelligent Systems (IMPRS-IS) for supporting JW and MP. JPC and GP are supported by the Simons Foundation, the McKnight Foundation, the Grossman Center, and the Gatsby Charitable Trust. The authors would like to thank Hanna Dettki, Frank Schneider, Lukas Tatzel and all anonymous reviewers for helpful feedback on an earlier version of this manuscript.

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
