# Supplementary Material:
# Posterior and Computational Uncertainty
# in Gaussian Processes

**Jonathan Wenger**[1,2]     **Geoff Pleiss**[2]

**Marvin Pförtner**[1]     **Philipp Hennig**[1,3]     **John P. Cunningham**[2]

[1] University of Tübingen
[2] Columbia University
[3] Max Planck Institute for Intelligent Systems, Tübingen

This supplementary material contains additional results and in particular proofs for all theoretical statements. References referring to sections, equations or theorem-type environments within this document are prefixed with 'S', while references to, or results from, the main paper are stated as is.

## S1  Connections to Other GP Approximations

### S1.1  Pivoted Cholesky Decomposition

**Theorem S3** (Pivoted Cholesky Decomposition)
*Let $(j_i)_{i=1}^{n}$ be a set of indices defining the pivot elements of the pivoted Cholesky decomposition and $\boldsymbol{P} \in \mathbb{R}^{n \times n}$ the corresponding permutation matrix. Assume the actions of Algorithm 1 are given by the standard unit vectors $\boldsymbol{s}_i = \boldsymbol{P}\boldsymbol{e}_i = \boldsymbol{e}_{j_i}$, i.e.*

$$(\boldsymbol{s}_i)_j = (\boldsymbol{e}_{j_i})_j = \begin{cases} 1 & \text{if } j = j_i \\ 0 & \text{otherwise} \end{cases}. \tag{S17}$$

*Then Algorithm 1 recovers the pivoted Cholesky decomposition, i.e. it holds for all $i \in \{0, \dots, n\}$ that*

$$\boldsymbol{P}^{\mathsf{T}}\boldsymbol{Q}_i\boldsymbol{P} = \boldsymbol{L}_i\boldsymbol{L}_i^{\mathsf{T}}, \tag{S18}$$

*where $\boldsymbol{L}_i \in \mathbb{R}^{n \times i}$ is the (partial) Cholesky factor of $\boldsymbol{P}^{\mathsf{T}}\hat{\boldsymbol{K}}\boldsymbol{P}$ as computed by Algorithm S2.*

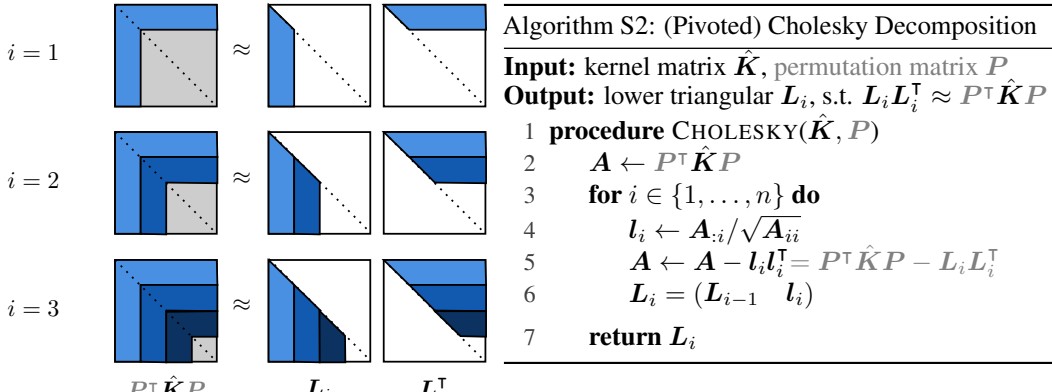

| | | | Algorithm S2: (Pivoted) Cholesky Decomposition |
|---|---|---|---|

**Input:** kernel matrix $\hat{K}$, permutation matrix $P$
**Output:** lower triangular $L_i$, s.t. $L_i L_i^\mathsf{T} \approx P^\mathsf{T} \hat{K} P$

1 **procedure** CHOLESKY($\hat{K}, P$)
2      $A \leftarrow P^\mathsf{T} \hat{K} P$
3      **for** $i \in \{1, \dots, n\}$ **do**
4          $l_i \leftarrow A_{:i}/\sqrt{A_{ii}}$
5          $A \leftarrow A - l_i l_i^\mathsf{T} = P^\mathsf{T} \hat{K} P - L_i L_i^\mathsf{T}$
6          $L_i = (L_{i-1} \quad l_i)$
7      **return** $L_i$

Figure S1: *Cholesky decomposition.* Every column added to the lower triangular Cholesky factor $L$ defines the $i$th "right angle ruler"-pattern in $P^\mathsf{T} \hat{K} P$. The bottom right matrix in gray given by $P^\mathsf{T} \hat{K} P - L_i L_i^\mathsf{T} = P^\mathsf{T} \hat{K} P - \sum_{j=1}^{i} l_j l_j^\mathsf{T}$ changes every iteration.

*Proof.* We proceed by induction. Assume (S18) holds after $i$ iterations of Algorithm 1. For the base case $i = 0$, it holds by assumption that $P^\mathsf{T} Q_0 P = P^\mathsf{T} \hat{K} C_0 \hat{K} P = 0$. Now for the induction step $i \to i + 1$, we have

$$
\begin{aligned}
\frac{1}{\eta_{i+1}} \hat{K} d_i d_i^\mathsf{T} \hat{K} &= \frac{1}{\eta_{i+1}} \hat{K} \Sigma_i \hat{K} s_{i+1} s_{i+1}^\mathsf{T} \hat{K} \Sigma_i \hat{K} \\
&= \frac{1}{\eta_{i+1}} \hat{K} (\Sigma_0 - C_i) \hat{K} s_{i+1} s_{i+1}^\mathsf{T} \hat{K} (\Sigma_0 - C_i) \hat{K} \\
&= \frac{1}{\eta_{i+1}} (\hat{K} - Q_i) s_{i+1} s_{i+1}^\mathsf{T} (\hat{K} - Q_i) \\
&\overset{\text{IH}}{=} \frac{1}{\eta_{i+1}} (\hat{K} - P L_i L_i^\mathsf{T} P^\mathsf{T}) s_{i+1} s_{i+1}^\mathsf{T} (\hat{K} - P L_i L_i^\mathsf{T} P^\mathsf{T}) \\
&= \frac{(\hat{K} - P L_i L_i^\mathsf{T} P^\mathsf{T}) P e_{i+1}}{\sqrt{e_{i+1}^\mathsf{T} P^\mathsf{T} (\hat{K} - P L_i L_i^\mathsf{T} P^\mathsf{T}) P e_{i+1}}} \frac{e_{i+1}^\mathsf{T} P^\mathsf{T} (\hat{K} - P L_i L_i^\mathsf{T} P^\mathsf{T})}{\sqrt{e_{i+1}^\mathsf{T} P^\mathsf{T} (\hat{K} - P L_i L_i^\mathsf{T} P^\mathsf{T}) P e_{i+1}}} \\
&= \frac{P (P^\mathsf{T} \hat{K} P - L_i L_i^\mathsf{T}) e_{i+1}}{\sqrt{e_{i+1}^\mathsf{T} (P^\mathsf{T} \hat{K} P - L_i L_i^\mathsf{T}) e_{i+1}}} \frac{e_{i+1}^\mathsf{T} (P^\mathsf{T} \hat{K} P - L_i L_i^\mathsf{T}) P^\mathsf{T}}{\sqrt{e_{i+1}^\mathsf{T} (P^\mathsf{T} \hat{K} P - L_i L_i^\mathsf{T}) e_{i+1}}} \\
&= P l_{i+1} l_{i+1}^\mathsf{T} P^\mathsf{T}.
\end{aligned}
$$

where $l_{i+1}$ is given by Algorithm S2. Combining this with one more use of the induction hypothesis we obtain

$$
\begin{aligned}
P^\mathsf{T} Q_{i+1} P &= P^\mathsf{T} Q_i P + \frac{1}{\eta_{i+1}} P^\mathsf{T} \hat{K} d_{i+1} d_{i+1}^\mathsf{T} \hat{K} P \\
&= L_i L_i^\mathsf{T} + l_{i+1} l_{i+1}^\mathsf{T} = (L_i \quad l_{i+1}) \begin{pmatrix} L_i^\mathsf{T} \\ l_{i+1}^\mathsf{T} \end{pmatrix} = L_{i+1} L_{i+1}^\mathsf{T}
\end{aligned}
$$

This proves the claim.

$\square$

## S1.2    Singular / Eigenvalue Decomposition

**Theorem S4** (Singular / Eigenvalue Decomposition)
*Let the actions $s_i = \mathbf{u}_i$ of Algorithm 1 be given by the eigenvectors $\mathbf{u}_i$ of $\hat{K}$ in arbitrary order. Then*

*for $i \in \{1, \ldots, n\}$ it holds that*

$$C_i = U_i \Lambda_i^{-1} U_i^\mathsf{T} = \mathrm{SVD}_i(\hat{K}^{-1})$$
$$Q_i = U_i \Lambda_i U_i^\mathsf{T} = \mathrm{SVD}_i(\hat{K}),$$

*where $U = (\mathbf{u}_1, \ldots, \mathbf{u}_i) \in \mathbb{R}^{n \times i}$ and $\Lambda = \mathrm{diag}(\lambda_1, \ldots, \lambda_i) \in \mathbb{R}^{i \times i}$ is the diagonal matrix of eigenvalues of $\hat{K}$ with the order given by the order of the actions.*

*Proof.* It holds by assumption and eq. (S37), that

$$C_i = S_i(S_i^\mathsf{T} \hat{K} S_i)^{-1} S_i^\mathsf{T} = U_i(U_i^\mathsf{T} \hat{K} U_i)^{-1} U_i^\mathsf{T} = U_i \Lambda_i^{-1} U_i^\mathsf{T},$$

as well as

$$Q_i = \hat{K} C_i \hat{K} = \hat{K} U_i \Lambda_i^{-1} U_i^\mathsf{T} \hat{K} = U_i \Lambda_i \Lambda_i^{-1} \Lambda_i U_i^\mathsf{T} = U_i \Lambda_i U_i^\mathsf{T}$$

This proves the claim. $\qquad\square$

## S1.3 Conjugate Gradient Method

---
Algorithm S3: Preconditioned Conjugate Gradient Method [38]

---
**Input:** kernel matrix $\hat{K}$, labels $y$, prior mean $\mu$, preconditioner $\hat{P}$
**Output:** representer weights $v_i \approx \hat{K}^{-1}(y - \mu)$

1    **procedure** CG($\hat{K}, y - \mu, \hat{P}$)
2       $v_0 \leftarrow 0$
3       $s_0 \leftarrow 0$
4       **while** $\|r_i\|_2 > \max(\delta_{\mathrm{rtol}}\|y\|_2, \delta_{\mathrm{atol}})$ **and** $i < i_{\max}$ **do**
5         $r_{i-1} \leftarrow (y - \mu) - \hat{K} v_{i-1}$
6         $s_i \leftarrow \hat{P}^{-1} r_{i-1} - \dfrac{(\hat{P}^{-1} r_{i-1})^\mathsf{T} \hat{K} s_{i-1}}{s_{i-1}^\mathsf{T} \hat{K} s_{i-1}} s_{i-1}$
7         $v_i \leftarrow v_{i-1} + \dfrac{(\hat{P}^{-1} r_{i-1})^\mathsf{T} r_{i-1}}{s_i^\mathsf{T} \hat{K} s_i} s_i$
8       **return** $v$

---

**Theorem S5** (Preconditioned Conjugate Gradient Method)
*Let $\hat{P} \in \mathbb{R}^{n \times n}$ be a symmetric positive definite preconditioner. Assume the actions of Algorithm 1 are given by*

$$s_1^{\mathrm{CG}} = \hat{P}^{-1} r_0$$
$$s_i^{\mathrm{CG}} = \hat{P}^{-1} r_{i-1} - \frac{(\hat{P}^{-1} r_{i-1})^\mathsf{T} \hat{K} s_{i-1}}{s_{i-1}^\mathsf{T} \hat{K} s_{i-1}} s_{i-1} \qquad (S19)$$

*the preconditioned conjugate gradient method. Then Algorithm 1 recovers preconditioned CG initialized at $v_0^{\mathrm{CG}} = 0$, i.e. it holds for $i \in \{1, \ldots, n\}$ that*

$$s_i = d_i = s_i^{\mathrm{CG}} \qquad (S20)$$
$$v_i = v_i^{\mathrm{CG}} \qquad (S21)$$
$$r_{i-1} = r_{i-1}^{\mathrm{CG}}. \qquad (S22)$$

*Proof.* First note that by assumption $s_i = s_i^{\mathrm{CG}}$ for all $i$. We prove the remaining claims by induction. For the base case we have by assumption $d_0 = \Sigma_0 \hat{K} s_0 = s_0 = s_0^{\mathrm{CG}}$ and $v_0 = 0 = v_0^{\mathrm{CG}}$. Now for the induction step $i \to i + 1$ assume the hypotheses (S20), (S21) and (S22) hold $\forall j \le i$. Using the properties of CG it holds for $j' < i$ that

$$s_i^\mathsf{T} \hat{K} s_{j'} = 0 \qquad (S23)$$
$$(\hat{P}^{-1} r_i)^\mathsf{T} s_{j'} = 0 \qquad (S24)$$
$$(\hat{P}^{-1} r_i)^\mathsf{T} r_{j'} = 0 \qquad (S25)$$
$$\langle s_1, \ldots, s_i \rangle = \langle r_0, \hat{P}^{-1} \hat{K} r_0, \ldots, (\hat{P}^{-1} \hat{K})^{i-1} r_0 \rangle = \langle \hat{P}^{-1} r_0, \ldots, \hat{P}^{-1} r_{i-1} \rangle \qquad (S26)$$

We now first show $\hat{\boldsymbol{K}}$-conjugacy of the actions in iteration $i+1$. We have for $j \leq i$ that

$$\boldsymbol{s}_{i+1}^{\mathsf{T}}\hat{\boldsymbol{K}}\boldsymbol{s}_j = \left(\hat{\boldsymbol{P}}^{-1}\boldsymbol{r}_i - \frac{(\hat{\boldsymbol{P}}^{-1}\boldsymbol{r}_i)^{\mathsf{T}}\hat{\boldsymbol{K}}\boldsymbol{s}_i}{\boldsymbol{s}_i^{\mathsf{T}}\hat{\boldsymbol{K}}\boldsymbol{s}_i}\boldsymbol{s}_i\right)^{\mathsf{T}}\hat{\boldsymbol{K}}\boldsymbol{s}_j$$

$$= (\hat{\boldsymbol{P}}^{-1}\boldsymbol{r}_i)^{\mathsf{T}}\hat{\boldsymbol{K}}\boldsymbol{s}_j - \frac{(\hat{\boldsymbol{P}}^{-1}\boldsymbol{r}_i)^{\mathsf{T}}\hat{\boldsymbol{K}}\boldsymbol{s}_i}{\boldsymbol{s}_i^{\mathsf{T}}\hat{\boldsymbol{K}}\boldsymbol{s}_i}\boldsymbol{s}_i^{\mathsf{T}}\hat{\boldsymbol{K}}\boldsymbol{s}_j$$

Now if $j = i$, clearly $\boldsymbol{s}_{i+1}^{\mathsf{T}}\hat{\boldsymbol{K}}\boldsymbol{s}_j = \boldsymbol{s}_{i+1}^{\mathsf{T}}\hat{\boldsymbol{K}}\boldsymbol{s}_i = 0$. If $j < i$, we have using (S26), that

$$\hat{\boldsymbol{P}}^{-1}\hat{\boldsymbol{K}}\boldsymbol{s}_j \in \langle \hat{\boldsymbol{P}}^{-1}\hat{\boldsymbol{K}}\boldsymbol{r}_0, \ldots, (\hat{\boldsymbol{P}}^{-1}\hat{\boldsymbol{K}})^j\boldsymbol{r}_0\rangle \subset \langle \hat{\boldsymbol{P}}^{-1}\boldsymbol{r}_0, \ldots, \hat{\boldsymbol{P}}^{-1}\boldsymbol{r}_j\rangle. \tag{S27}$$

Therefore we obtain for $j < i$, that

$$\boldsymbol{s}_{i+1}^{\mathsf{T}}\hat{\boldsymbol{K}}\boldsymbol{s}_j \overset{\text{(S23)}}{=} \boldsymbol{r}_i^{\mathsf{T}}\hat{\boldsymbol{P}}^{-1}\hat{\boldsymbol{K}}\boldsymbol{s}_j \overset{\text{(S27)}}{=} \boldsymbol{r}_i^{\mathsf{T}}\left(\sum_{\ell=1}^{j}\gamma_\ell\hat{\boldsymbol{P}}^{-1}\boldsymbol{r}_\ell\right) \overset{\text{(S25)}}{=} 0. \tag{S28}$$

Thus in combination we have

$$\forall j \in \{1, \ldots, i\}: \quad \boldsymbol{s}_{i+1}^{\mathsf{T}}\hat{\boldsymbol{K}}\boldsymbol{s}_j = 0. \tag{S29}$$

Now for the search direction we have

$$\boldsymbol{d}_{i+1} = \boldsymbol{\Sigma}_i\hat{\boldsymbol{K}}\boldsymbol{s}_{i+1} = \left(\boldsymbol{\Sigma}_0 - \sum_{j=1}^{i}\frac{\boldsymbol{d}_j\boldsymbol{d}_j^{\mathsf{T}}}{\eta_j}\right)\hat{\boldsymbol{K}}\boldsymbol{s}_{i+1}$$

$$= \boldsymbol{s}_{i+1} - \sum_{j=1}^{i}\frac{\boldsymbol{d}_j^{\mathsf{T}}\hat{\boldsymbol{K}}\boldsymbol{s}_{i+1}}{\eta_j}\boldsymbol{d}_j \overset{\text{(S20)}}{=} \boldsymbol{s}_{i+1} - \sum_{j=1}^{i}\frac{\boldsymbol{s}_j^{\mathsf{T}}\hat{\boldsymbol{K}}\boldsymbol{s}_{i+1}}{\eta_j}\boldsymbol{d}_j \tag{S30}$$

$$\overset{\text{(S29)}}{=} \boldsymbol{s}_{i+1}.$$

Further, we have for the solution estimate, that $\boldsymbol{v}_{i+1} = \boldsymbol{v}_i + \boldsymbol{d}_{i+1}\frac{\alpha_{i+1}}{\eta_{i+1}}$. It holds that

$$\alpha_{i+1} = \boldsymbol{s}_{i+1}^{\mathsf{T}}\boldsymbol{r}_i = \left(\hat{\boldsymbol{P}}^{-1}\boldsymbol{r}_i - \frac{(\hat{\boldsymbol{P}}^{-1}\boldsymbol{r}_i)^{\mathsf{T}}\hat{\boldsymbol{K}}\boldsymbol{s}_i}{\boldsymbol{s}_i^{\mathsf{T}}\hat{\boldsymbol{K}}\boldsymbol{s}_i}\boldsymbol{s}_i\right)^{\mathsf{T}}\boldsymbol{r}_i$$

$$= (\hat{\boldsymbol{P}}^{-1}\boldsymbol{r}_i)^{\mathsf{T}}\boldsymbol{r}_i - \sum_{j=}^{i}c_j(\hat{\boldsymbol{P}}^{-1}\boldsymbol{r}_{j-1})^{\mathsf{T}}\boldsymbol{r}_i \overset{\text{(S25)}}{=} (\hat{\boldsymbol{P}}^{-1}\boldsymbol{r}_i)^{\mathsf{T}}\boldsymbol{r}_i$$

as well as

$$\eta_{i+1} = \boldsymbol{s}_{i+1}^{\mathsf{T}}\hat{\boldsymbol{K}}\boldsymbol{\Sigma}_i\hat{\boldsymbol{K}}\boldsymbol{s}_{i+1} = \boldsymbol{d}_{i+1}^{\mathsf{T}}\hat{\boldsymbol{K}}\boldsymbol{s}_{i+1} \overset{\text{(S30)}}{=} \boldsymbol{s}_{i+1}^{\mathsf{T}}\hat{\boldsymbol{K}}\boldsymbol{s}_{i+1}$$

Combining the above and recalling Algorithm S3, we obtain

$$\boldsymbol{v}_{i+1} = \boldsymbol{v}_i + \boldsymbol{d}_{i+1}\frac{\alpha_{i+1}}{\eta_{i+1}} = \boldsymbol{v}_i + \boldsymbol{d}_{i+1}\frac{(\hat{\boldsymbol{P}}^{-1}\boldsymbol{r}_i)^{\mathsf{T}}\boldsymbol{r}_i}{\boldsymbol{s}_{i+1}^{\mathsf{T}}\hat{\boldsymbol{K}}\boldsymbol{s}_{i+1}} = \boldsymbol{v}_{i+1}^{\text{CG}}.$$

Finally, the residual is computed identically in Algorithm 1 as in Algorithm S3, giving

$$\boldsymbol{r}_i = (\boldsymbol{y} - \boldsymbol{\mu}) - \hat{\boldsymbol{K}}\boldsymbol{v}_i = (\boldsymbol{y} - \boldsymbol{\mu}) - \hat{\boldsymbol{K}}\boldsymbol{v}_i^{\text{CG}} = \boldsymbol{r}_i^{\text{CG}}.$$

This proves the claims. $\square$

**Corollary S2** (Preconditioned Gradient Actions as CG Actions)
*Choosing actions*

$$\boldsymbol{s}_i = \hat{\boldsymbol{P}}^{-1}\boldsymbol{r}_{i-1} \tag{S31}$$

*in Theorem S5 instead also reproduces the preconditioned conjugate gradient method, i.e. it holds for $i \in \{1, \ldots, n\}$ that*

$$\boldsymbol{d}_i = \boldsymbol{s}_i^{\text{CG}} \tag{S32}$$

$$\boldsymbol{v}_i = \boldsymbol{v}_i^{\text{CG}} \tag{S33}$$

$$\boldsymbol{r}_{i-1} = \boldsymbol{r}_{i-1}^{\text{CG}}. \tag{S34}$$

*Proof.* It suffices to show that $d_i = s_i^{\mathrm{CG}}$. The rest of the argument is then identical to the proof of Theorem S5. We prove the claim by induction. For the base case by assumption $s_1 = \hat{P}^{-1} r_0 = s_1^{\mathrm{CG}}$. Now for the induction step $i \to i+1$, assume that $d_j = s_j$ for all $j \leq i$, then

$$
\begin{aligned}
d_{i+1} &= \Sigma_i \hat{K} \hat{P}^{-1} r_i \\
&= (I - C_i \hat{K}) \hat{P}^{-1} r_i \\
&= \hat{P}^{-1} r_i - D_i (D_i^\mathsf{T} \hat{K} D_i)^{-1} D_i^\mathsf{T} \hat{K} \hat{P}^{-1} r_i && \text{By eq. (S37).} \\
&\overset{\mathrm{IH}}{=} \hat{P}^{-1} r_i - S_i^{\mathrm{CG}} ((S_i^{\mathrm{CG}})^\mathsf{T} \hat{K} S_i^{\mathrm{CG}})^{-1} (S_i^{\mathrm{CG}})^\mathsf{T} \hat{K} \hat{P}^{-1} r_i
\end{aligned}
$$

Now by the same argument as in eq. (S28) in the proof of Theorem S5 we have for all $j < i$ that $r_i^\mathsf{T} \hat{P}^{-1} \hat{K} s_j^{\mathrm{CG}} = 0$. Therefore

$$
\begin{aligned}
&= \hat{P}^{-1} r_i - s_i^{\mathrm{CG}} ((s_i^{\mathrm{CG}})^\mathsf{T} \hat{K} s_i^{\mathrm{CG}})^{-1} (s_i^{\mathrm{CG}})^\mathsf{T} \hat{K} \hat{P}^{-1} r_i \\
&= s_{i+1}^{\mathrm{CG}} && \text{By eq. (S19).}
\end{aligned}
$$

This proves the claim. $\qquad\square$

**Corollary S3** (Deflated Conjugate Gradient Method)
*Let the first $0 < \ell < n$ actions $(s_i)_{i=1}^\ell$ of Algorithm 1 be linearly independent and the remaining ones be given by $s_i = \hat{P}^{-1} r_i$, where $\hat{P} \approx \hat{K}$ is a preconditioner. Then Algorithm 1 is equivalent to the preconditioned deflated CG algorithm [63, Alg. 3.6] with deflation subspace $\mathrm{span}\{S_\ell\}$.*

*Proof.* By the form of preconditioned deflated CG given in Algorithm 3.6 of Saad et al. [63] and Corollary S2, it suffices to show that the residual $r_\ell$ satisfies $S_\ell^\mathsf{T} r_\ell = 0$ and that for all $i > \ell$, it holds that
$$
s_i^{\mathrm{defCG}} = d_i = (I - C_{i-1} \hat{K}) s_i.
$$
Now it holds by Lemma S2 and eq. (S37), that
$$
S_\ell^\mathsf{T} r_\ell = S_\ell^\mathsf{T} (I - \hat{K} C_\ell)(y - \mu) = \underbrace{S_\ell^\mathsf{T} (I - \hat{K} S_\ell (S_\ell^\mathsf{T} \hat{K} S_\ell)^{-1} S_\ell^\mathsf{T})}_{=0}(y - \mu) = 0.
$$

This proves the first claim. Now, by Saad et al. [63, Alg. 3.6], the search directions $(s_i^{\mathrm{defCG}})_{i=\ell+1}^n$ of preconditioned deflated CG are given by

$$
\begin{aligned}
s_i^{\mathrm{defCG}} &= s_i^{\mathrm{CG}} - S_\ell (S_\ell^\mathsf{T} \hat{K} S_\ell)^{-1} S_\ell^\mathsf{T} \hat{K} \hat{P}^{-1} r_i \\
&= (I - C_{\ell+1:(i-1)} \hat{K}) s_i - S_\ell (S_\ell^\mathsf{T} \hat{K} S_\ell)^{-1} S_\ell^\mathsf{T} \hat{K} \hat{P}^{-1} r_i && \text{Corollary S2} \\
&= (I - C_{\ell+1:(i-1)} \hat{K}) s_i - C_\ell \hat{K} s_i \\
&= (I - (C_{\ell+1:(i-1)} - C_\ell) \hat{K}) s_i \\
&= (I - C_{i-1} \hat{K}) s_i \\
&= d_i
\end{aligned}
$$

This proves the claim. $\qquad\square$

**Remark S1** (Preconditioning and Algorithm 1)
Iterative methods typically have convergence rates depending on the condition number of the system matrix. One successful strategy in practice to accelerate convergence is to use a preconditioner $\hat{P} \approx \hat{K}$ [64]. A preconditioner needs to be cheap to compute and allow efficient matrix-vector multiplication $v \mapsto \hat{P}^{-1} v$. Now, Algorithm 1 implicitly constructs and applies a *deflation-based preconditioner*, which are defined via a deflation subspace to be projected out [65]. In Algorithm 1 this is precisely the already explored space $\mathrm{span}\{S_i\} = \mathrm{span}\{D_i\}$ spanned by the actions. Therefore, if we run a mixed strategy, meaning first choosing actions that define a certain subspace and then choose residual actions, we recover the *deflated conjugate gradient method* [63] (see Corollary S3 for a proof). Alternatively, one can also use byproducts of the iteration of Algorithm 1 to construct a diagonal-plus-low-rank preconditioner of the form $\hat{P} = \sigma^2 I + U U^\mathsf{T} \approx \hat{K}$ where $U = K D_i \mathrm{diag}(\eta_1, \ldots, \eta_i) \in \mathbb{R}^{n \times i}$. Therefore, again if running a mixed strategy, one can first construct a preconditioner and then accelerate the convergence of subsequent CG iterations. In this sense one can double-dip in terms of preconditioning (conjugate) gradient iterations by combining these two techniques *at essentially no overhead*.

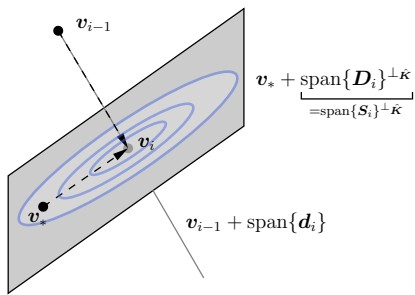

Figure S2: *Geometric perspective on the probabilistic linear solver learning representer weights $\boldsymbol{v}_*$.*

### S1.4 Inducing Point Methods

**Theorem S6** (Approximate Posterior Mean of Nyström, SoR, DTC and SVGP)
*Let $\boldsymbol{Z} \in \mathbb{R}^{n \times m}$ be a set of distinct inducing inputs such that $\mathrm{rank}(\boldsymbol{K_{XZ}}) = m \leq n$. Then the posterior mean of the Nyström variants subset of regressors (SoR) and deterministic training conditional (DTC) is identical to the one of SVGP and given by*

$$
\begin{aligned}
\mu(\cdot) &= k(\cdot, \boldsymbol{Z})(\boldsymbol{K_{ZX}K_{XZ}} + \sigma^2 \boldsymbol{K_{ZZ}})^{-1} \boldsymbol{K_{ZX}}(\boldsymbol{y} - \boldsymbol{\mu}) \\
&= q(\cdot, \boldsymbol{X})\boldsymbol{K_{XZ}}(\boldsymbol{K_{ZX}}(q(\boldsymbol{X}, \boldsymbol{X}) + \sigma^2 \boldsymbol{I})\boldsymbol{K_{XZ}})^{-1}\boldsymbol{K_{ZX}}(\boldsymbol{y} - \boldsymbol{\mu})
\end{aligned}
\tag{S35}
$$

*Proof.* First, note that by eqns. (16b) and (20b) of Quiñonero-Candela and Rasmussen [20] the posterior mean of SoR and DTC is identical and given by

$$
\mu(\cdot) = k(\cdot, \boldsymbol{Z})(\boldsymbol{K_{ZX}K_{XZ}} + \sigma^2 \boldsymbol{K_{ZZ}})^{-1} \boldsymbol{K_{ZX}}(\boldsymbol{y} - \boldsymbol{\mu})
$$

Now, by Theorem 5 of Wild et al. [43] the posterior mean of SVGP for a fixed set of inducing points is equivalent to the Nyström approximation, which takes the form above. Recognizing that $\boldsymbol{K_{ZX}K_{XZ}} \in \mathbb{R}^{m \times m}$ is invertible, it holds that

$$
\begin{aligned}
\mu(\cdot) &= k(\cdot, \boldsymbol{Z})(\boldsymbol{K_{ZX}K_{XZ}} + \sigma^2 \boldsymbol{K_{ZZ}})^{-1} \boldsymbol{K_{ZX}}(\boldsymbol{y} - \boldsymbol{\mu}) \\
&= k(\cdot, \boldsymbol{Z})(\boldsymbol{K_{ZZ}}(\boldsymbol{K_{ZZ}^{-1}K_{ZX}K_{XZ}} + \sigma^2 \boldsymbol{I}))^{-1} \boldsymbol{K_{ZX}}(\boldsymbol{y} - \boldsymbol{\mu}) \\
&= k(\cdot, \boldsymbol{Z})\boldsymbol{K_{ZZ}^{-1}}((\boldsymbol{K_{ZX}K_{XZ}})^{-1}(\boldsymbol{K_{ZX}K_{XZ}K_{ZZ}^{-1}K_{ZX}K_{XZ}} + \sigma^2 \boldsymbol{K_{ZX}K_{XZ}}))^{-1} \boldsymbol{K_{ZX}}(\boldsymbol{y} - \boldsymbol{\mu}) \\
&= k(\cdot, \boldsymbol{Z})\boldsymbol{K_{ZZ}^{-1}K_{ZX}K_{XZ}}(\boldsymbol{K_{ZX}}(\boldsymbol{K_{XZ}K_{ZZ}^{-1}K_{ZX}} + \sigma^2 \boldsymbol{I})\boldsymbol{K_{XZ}})^{-1} \boldsymbol{K_{ZX}}(\boldsymbol{y} - \boldsymbol{\mu}) \\
&= q(\cdot, \boldsymbol{X})\boldsymbol{K_{XZ}}(\boldsymbol{K_{ZX}}(q(\boldsymbol{X}, \boldsymbol{X}) + \sigma^2 \boldsymbol{I})\boldsymbol{K_{XZ}})^{-1}\boldsymbol{K_{ZX}}(\boldsymbol{y} - \boldsymbol{\mu})
\end{aligned}
$$

This proves the claim. $\qquad\square$

## S2 Theoretical Results and Proofs

### S2.1 Properties of Algorithm 1

**Lemma S1** (Geometric Properties of Algorithm 1)
*Let $i \in \{1, \dots, n\}$, and assume $\boldsymbol{\Sigma}_0$ is chosen such that $\boldsymbol{\Sigma}_0 \hat{\boldsymbol{K}} \boldsymbol{s}_j = \boldsymbol{s}_j$ for all $j \leq i$ (e.g. $\boldsymbol{\Sigma}_0 = \hat{\boldsymbol{K}}^{-1}$). Then it holds for the quantities computed by Algorithm 1 that*

$$
\mathrm{span}\{\boldsymbol{S}_i\} = \mathrm{span}\{\boldsymbol{D}_i\}
\tag{S36}
$$

$$
\boldsymbol{C}_i = \boldsymbol{D}_i(\boldsymbol{D}_i^\mathsf{T}\hat{\boldsymbol{K}}\boldsymbol{D}_i)^{-1}\boldsymbol{D}_i^\mathsf{T} = \boldsymbol{S}_i(\boldsymbol{S}_i^\mathsf{T}\hat{\boldsymbol{K}}\boldsymbol{S}_i)^{-1}\boldsymbol{S}_i^\mathsf{T}
\tag{S37}
$$

$$
\boldsymbol{C}_i\hat{\boldsymbol{K}} \text{ is the } \hat{\boldsymbol{K}}\text{-orthogonal projection onto } \mathrm{span}\{\boldsymbol{D}_i\}
\tag{S38}
$$

$$
\boldsymbol{\Sigma}_i\hat{\boldsymbol{K}} \text{ is the } \hat{\boldsymbol{K}}\text{-orthogonal projection onto } \mathrm{span}\{\boldsymbol{D}_i\}^{\perp \hat{K}}
\tag{S39}
$$

$$
\boldsymbol{d}_i^\mathsf{T}\hat{\boldsymbol{K}}\boldsymbol{d}_j = 0 \qquad \text{for all } j < i
\tag{S40}
$$

*where $\boldsymbol{S}_i = (\boldsymbol{s}_1 \cdots \boldsymbol{s}_i) \in \mathbb{R}^{n \times i}$ and $\boldsymbol{D}_i = (\boldsymbol{d}_1 \cdots \boldsymbol{d}_i) \in \mathbb{R}^{n \times i}$.*

*Proof.* We prove the claims by induction. We begin with the base case $i = 1$.

By assumption it holds that $S_1 = s_1 = \Sigma_0 \hat{K} s_1 = d_1 = D_1$. Now by Algorithm 1, we have $C_1 = \frac{1}{\eta_1} d_1 d_1^\mathsf{T}$, which with the above proves (S37). By the batched form (S37) of $C_i$, the statements (S38) and (S39) follow immediately. Finally, $\hat{K}$-orthogonality for a single search direction holds trivially.

Now for the induction step $i \to i + 1$. Assume that eqs. (S36) to (S40) hold for iteration $i$. Then we have that

$$d_{i+1} = \Sigma_i \hat{K} s_{i+1} = s_{i+1} - C_i \hat{K} s_{i+1} \overset{(S37)}{=} s_{i+1} - S_i (S_i^\mathsf{T} \hat{K} S_i)^{-1} S_i^\mathsf{T} \hat{K} s_{i+1} \in \mathrm{span}\{S_{i+1}\}$$

By the induction hypothesis the above also implies $\mathrm{span}\{S_{i+1}\} = \mathrm{span}\{D_{i+1}\}$. This proves eq. (S36). Next, we have by the induction hypotheses (S37) and (S40) that

$$C_{i+1} = C_i + \frac{1}{\eta} d_{i+1} d_{i+1}^\mathsf{T}$$

$$= D_i (D_i^\mathsf{T} \hat{K} D_i)^{-1} D_i^\mathsf{T} + \frac{1}{\eta_{i+1}} d_{i+1} d_{i+1}^\mathsf{T}$$

$$= \sum_{k=1}^{i+1} \frac{1}{\eta_k} d_k d_k^\mathsf{T}$$

$$= D_{i+1} (D_{i+1}^\mathsf{T} \hat{K} D_{i+1})^{-1} D_{i+1}^\mathsf{T}$$

This proves the first equality of eq. (S37). For the second, first recognize that an orthogonal projection onto a linear subspace $\mathrm{span}\{A\}$ with respect to the $B$-inner product is given by $P_A = A(A^\mathsf{T} B A)^{-1} A^\mathsf{T} B$. The projection onto its $B$-orthogonal subspace is given by $P_{A^\perp B} = I - P_A$. Therefore eqs. (S38) and (S39) follow directly from the above argument. Now since projection onto a subspace is unique and independent of the choice of basis, we have by $\mathrm{span}\{D_{i+1}\} = \mathrm{span}\{S_{i+1}\}$ that

$$C_i \hat{K} = P_{D_{i+1}} = P_{S_{i+1}} = S_i (S_i^\mathsf{T} \hat{K} S_i)^{-1} S_i^\mathsf{T} \hat{K}$$

Now since $\hat{K}$ is non-singular, the second equality of eq. (S37) follows. Finally, we will prove $\hat{K}$-orthogonality of the search directions. Let $j < i + 1$, then it holds that

$$d_{i+1}^\mathsf{T} \hat{K} d_j = (\underbrace{\Sigma_i \hat{K} s_{i+1}}_{\in \mathrm{span}\{S_i\}^{\perp \hat{K}}})^\mathsf{T} \hat{K} \underbrace{d_j}_{\in \mathrm{span}\{S_i\}} = 0$$

by eqs. (S36) and (S39). This completes the proof. $\qquad \square$

**Corollary S4**
*Let $i \in \{1, \ldots, n\}$. It holds for $C_i \hat{K}$, the $\hat{K}$-orthogonal projection onto $S_i$, that*

$$(C_i \hat{K})^2 = C_i \hat{K} \tag{S41}$$

$$C_i \hat{K} C_i = C_i \tag{S42}$$

*Further for $H_i = \Sigma_i \hat{K} = I - C_i \hat{K}$ the $\hat{K}$-orthogonal projection onto $S_i^{\perp \hat{K}}$, we have*

$$H_i^2 = H_i \tag{S43}$$

$$H_i^\mathsf{T} \hat{K} H_i = H_i^\mathsf{T} \hat{K} = \hat{K} H_i \tag{S44}$$

*Proof.* By Lemma S1, it holds that $C_i = S_i (S_i^\mathsf{T} \hat{K} S_i)^{-1} S_i^\mathsf{T}$. Therefore

$$C_i \hat{K} C_i = S_i (S_i^\mathsf{T} \hat{K} S_i)^{-1} S_i^\mathsf{T} \hat{K} S_i (S_i^\mathsf{T} \hat{K} S_i)^{-1} S_i^\mathsf{T} = C_i.$$

This proves (S42) and (S41). Define $H_i = I - C_i \hat{K}$, then

$$H_i H_i = (I - C_i \hat{K})(I - C_i \hat{K}) = I - 2C_i \hat{K} + (C_i \hat{K})^2 = I - C_i \hat{K} = H_i$$

as well as

$$H_i^\mathsf{T} \hat{K} H_i = (I - C_i \hat{K})^\mathsf{T} \hat{K} (I - C_i \hat{K}) = (\hat{K} - \hat{K} C_i \hat{K})(I - C_i \hat{K})$$

$$= \hat{K} - 2\hat{K} C_i \hat{K} + \hat{K} (C_i \hat{K})^2$$

$$= \hat{K} - \hat{K} C_i \hat{K} = H_i^\mathsf{T} \hat{K} = \hat{K} H_i.$$

$\qquad \square$

**Lemma S2**
Let $\Sigma_0 = \hat{K}^{-1}$, then it holds that

$$C_i(y - \mu) = v_i, \tag{S45}$$
$$\Sigma_i(y - \mu) = v_* - v_i. \tag{S46}$$

*Proof.* We prove the statement by induction. By assumption $C_0(y - \mu) = v_0$. Now assume (S45) holds. Then for $i \to i + 1$, we have

$$C_{i+1}(y - \mu) = (C_i + \frac{1}{\eta_{i+1}} d_{i+1} d_{i+1}^\mathsf{T})(y - \mu) \overset{\text{IH}}{=} v_i + \frac{d_{i+1}^\mathsf{T}(y - \mu)}{\eta_{i+1}} d_{i+1}$$

Now by the update to the representer weights in Algorithm 1 it suffices to show that $\alpha_{i+1} = d_{i+1}^\mathsf{T}(y - \mu)$. We have

$$d_{i+1}^\mathsf{T}(y - \mu) = (\Sigma_i \hat{K} s_{i+1})^\mathsf{T}(y - \mu) = s_{i+1}^\mathsf{T} \hat{K} \Sigma_i (y - \mu)$$
$$= s_{i+1}^\mathsf{T} \hat{K}(\hat{K}^{-1} - C_i)(y - \mu) \overset{\text{IH}}{=} s_{i+1}^\mathsf{T}((y - \mu) - \hat{K} v_i) = s_{i+1}^\mathsf{T} r_i = \alpha_i.$$

$\square$

**Lemma S3**
Let $\Sigma_0 = \hat{K}^{-1}$, $C_0 = 0$ and consequently $v_0 = 0$, then it holds for the residual at iteration $i \in \{1, \dots, n\}$ that

$$r_{i-1} = \hat{K}(v_* - v_{i-1}) \tag{S47}$$
$$= \hat{K}\Sigma_{i-1}\hat{K}v_* \tag{S48}$$
$$= (\hat{K} - Q_{i-1})v_*. \tag{S49}$$

*Proof.* It holds by definition, that

$$r_{i-1} = (y - \mu) - \hat{K}v_{i-1} = \hat{K}v_* - \hat{K}v_{i-1} = \hat{K}(v_* - v_{i-1}).$$

Further we have by eq. (S46), that

$$= \hat{K}\Sigma_{i-1}(y - \mu) = \hat{K}\Sigma_{i-1}\hat{K}v_*,$$

and finally, by the definition of the kernel matrix approximation in Algorithm 1, we obtain

$$= \hat{K}(\hat{K}^{-1} - C_{i-1})\hat{K}v_* = (\hat{K} - Q_{i-1})v_*.$$

$\square$

**Proposition S3** (Batch of Observations)
Let $\Sigma_0$ such that $\Sigma_0 \hat{K} s_j = s_j$ for all $j \in \{1, \dots, i\}$. Then after $i$ iterations the posterior over the representer weights in (4) is equivalent to the one computed for a batch of observations, i.e.

$$v_i = \Sigma_0 \hat{K} S_i (S_i^\mathsf{T} \hat{K} \Sigma_0 \hat{K} S_i)^{-1} S_i^\mathsf{T}(y - \mu)$$
$$\Sigma_i = \Sigma_0 - \Sigma_0 \hat{K} S_i (S_i^\mathsf{T} \hat{K} \Sigma_0 \hat{K} S_i)^{-1} S_i^\mathsf{T} \hat{K} \Sigma_0$$

*Proof.* This can be seen as a direct consequence of recursively applying Bayes' theorem

$$p(v_* \mid \{\alpha_i\}_{i=1}^m, \{s_i\}_{i=1}^m) = \frac{p(\alpha_m \mid s_m, v_*)p(v_* \mid \{\alpha_i\}_{i=1}^{m-1}, \{s_i\}_{i=1}^{m-1})}{\int p(\alpha_m \mid s_m, v_*)p(v_* \mid \{\alpha_i\}_{i=1}^{m-1}, \{s_i\}_{i=1}^{m-1})dv_*}.$$

However, here we also give a geometric proof based on the projection property of the precision matrix approximation $C_i$. By using eq. (S37) and the assumption on $\Sigma_0$ we have that

$$C_i = S_i(S_i^\mathsf{T} \hat{K} S_i)^{-1} S_i^\mathsf{T} = \Sigma_0 \hat{K} S_i (S_i^\mathsf{T} \hat{K} \Sigma_0 \hat{K} S_i)^{-1} S_i^\mathsf{T}$$
$$= \Sigma_0 \hat{K} S_i (S_i^\mathsf{T} \hat{K} \Sigma_0 \hat{K} S_i)^{-1} S_i^\mathsf{T} \hat{K} \Sigma_0$$

This proves that

$$\Sigma_i = \Sigma_0 - C_i = \Sigma_0 - \Sigma_0 \hat{K} S_i (S_i^\mathsf{T} \hat{K} \Sigma_0 \hat{K} S_i)^{-1} S_i^\mathsf{T} \hat{K} \Sigma_0$$

Now by eq. (S45) it holds that $C_i(y - \mu) = v_i$. This proves the claim. $\square$

**Proposition S4** (Posterior Contraction)

*Let $\boldsymbol{S}_i \in \mathbb{R}^{n \times i}$ be the actions chosen by Algorithm 1, then its posterior contracts as*

$$\operatorname{tr}\big(\boldsymbol{\Sigma}_i \boldsymbol{\Sigma}_0^{-1}\big) = \operatorname{tr}(\boldsymbol{\Sigma}_i \hat{\boldsymbol{K}}) = n - \operatorname{rank}(\boldsymbol{S}_i).$$

*Proof.* Since $\boldsymbol{\Sigma}_0 = \hat{\boldsymbol{K}}^{-1}$, we have by eq. (S37), that

$$\begin{aligned}
\operatorname{tr}\big(\boldsymbol{\Sigma}_i \boldsymbol{\Sigma}_0^{-1}\big) &= \operatorname{tr}((\boldsymbol{\Sigma}_0 - \boldsymbol{C}_i)\hat{\boldsymbol{K}}) \\
&= \operatorname{tr}\big(\boldsymbol{I}_n - \boldsymbol{S}_i(\boldsymbol{S}_i^\mathsf{T} \hat{\boldsymbol{K}} \boldsymbol{S}_i)^\dagger \boldsymbol{S}_i^\mathsf{T} \hat{\boldsymbol{K}}\big) \\
&= \operatorname{tr}(\boldsymbol{I}_n) - \operatorname{tr}(\underbrace{\boldsymbol{S}_i^\mathsf{T} \hat{\boldsymbol{K}} \boldsymbol{S}_i(\boldsymbol{S}_i^\mathsf{T} \hat{\boldsymbol{K}} \boldsymbol{S}_i)^\dagger}_{\in \mathbb{R}^{i \times i}}) \\
&= n - \operatorname{rank}(\boldsymbol{S}_i)
\end{aligned}$$

Now, if the actions $\boldsymbol{S}_i$ are chosen linearly independent, then $\operatorname{rank}(\boldsymbol{S}_i) = i$. $\qquad\square$

**Theorem S7** (Online GP Approximation with Algorithm 1)

*Let $n, n' \in \mathbb{N}$ and consider training data sets $\boldsymbol{X} \in \mathbb{R}^{n \times d}, \boldsymbol{y} \in \mathbb{R}^n$ and $\boldsymbol{X}' \in \mathbb{R}^{n' \times d}, \boldsymbol{y}' \in \mathbb{R}^{n'}$. Consider two sequences of actions $(\boldsymbol{s}_i)_{i=1}^n \in \mathbb{R}^n$ and $(\tilde{\boldsymbol{s}}_i)_{i=1}^{n+n'} \in \mathbb{R}^{n+n'}$ such that for all $i \in \{1, \ldots, n\}$, it holds that*

$$\tilde{\boldsymbol{s}}_i = \begin{pmatrix} \boldsymbol{s}_i \\ \boldsymbol{0} \end{pmatrix} \tag{S50}$$

*Then the posterior returned by Algorithm 1 for the dataset $(\boldsymbol{X}, \boldsymbol{y})$ using actions $\boldsymbol{s}_i$ is identical to the posterior returned by Algorithm 1 for the extended dataset using actions $\tilde{\boldsymbol{s}}_i$, i.e. it holds for any $i \in \{1, \ldots, n\}$, that*

$$\operatorname{ITERGP}(\mu, k, \boldsymbol{X}, \boldsymbol{y}, (\boldsymbol{s}_i)_i) = (\mu_i, k_i) = (\tilde{\mu}_i, \tilde{k}_i) = \operatorname{ITERGP}\left(\mu, k, \begin{pmatrix} \boldsymbol{X} \\ \boldsymbol{X}' \end{pmatrix}, \begin{pmatrix} \boldsymbol{y} \\ \boldsymbol{y}' \end{pmatrix}, (\tilde{\boldsymbol{s}}_i)_i\right).$$

*Proof.* Define $\tilde{\boldsymbol{X}} = \begin{pmatrix} \boldsymbol{X} \\ \boldsymbol{X}' \end{pmatrix}$ and $\tilde{\boldsymbol{y}} = \begin{pmatrix} \boldsymbol{y} \\ \boldsymbol{y}' \end{pmatrix}$. We begin by showing that the search directions of both methods satisfy

$$\boldsymbol{d}_i' = \begin{pmatrix} \boldsymbol{d}_i \\ \boldsymbol{0} \end{pmatrix}. \tag{S51}$$

We proceed by induction. For $i = 0$ it holds by definition of Algorithm 1 and eq. (S50) that

$$\tilde{\boldsymbol{d}}_0 = \tilde{\boldsymbol{s}}_0 = \begin{pmatrix} \boldsymbol{s}_0 \\ \boldsymbol{0} \end{pmatrix} = \begin{pmatrix} \boldsymbol{d}_0 \\ \boldsymbol{0} \end{pmatrix}. \tag{S52}$$

Now for the induction step $i \to i+1$, assume that (S51) holds for $j \in \{1, \ldots, i\}$. Then, we have

$$\begin{aligned}
\tilde{\boldsymbol{d}}_{i+1} &= \tilde{\boldsymbol{\Sigma}}_{i-1}(k(\tilde{\boldsymbol{X}}, \tilde{\boldsymbol{X}}) + \sigma^2 \boldsymbol{I}_{n+n'})\tilde{\boldsymbol{s}}_{i+1} \\
&= (\boldsymbol{I}_{n+n'} - \tilde{\boldsymbol{C}}_i(k(\tilde{\boldsymbol{X}}, \tilde{\boldsymbol{X}}) + \sigma^2 \boldsymbol{I}_{n+n'}))\tilde{\boldsymbol{s}}_{i+1} \\
&= \tilde{\boldsymbol{s}}_{i+1} - \sum_{j=1}^i \frac{1}{\tilde{\eta}_j} \tilde{\boldsymbol{d}}_j (\tilde{\boldsymbol{d}}_j)^\mathsf{T} (k(\tilde{\boldsymbol{X}}, \tilde{\boldsymbol{X}}) + \sigma^2 \boldsymbol{I}_{n+n'})\tilde{\boldsymbol{s}}_{i+1} \\
&\overset{\text{IH}}{=} \begin{pmatrix} \boldsymbol{s}_{i+1} \\ \boldsymbol{0} \end{pmatrix} - \sum_{j=1}^i \frac{1}{\tilde{\eta}_j} \begin{pmatrix} \boldsymbol{d}_j \\ \boldsymbol{0} \end{pmatrix} \begin{pmatrix} \boldsymbol{d}_j^\mathsf{T} & \boldsymbol{0} \end{pmatrix} \begin{pmatrix} k(\boldsymbol{X}, \boldsymbol{X}) + \boldsymbol{I}_n & k(\boldsymbol{X}, \boldsymbol{X}') \\ k(\boldsymbol{X}', \boldsymbol{X}) & k(\boldsymbol{X}', \boldsymbol{X}') + \boldsymbol{I}_{n'} \end{pmatrix} \begin{pmatrix} \boldsymbol{s}_{i+1} \\ \boldsymbol{0} \end{pmatrix} \\
&= \begin{pmatrix} \boldsymbol{s}_{i+1} - \sum_{j=1}^i \frac{1}{\eta_j} \boldsymbol{d}_j (\boldsymbol{d}_j)^\mathsf{T} \hat{\boldsymbol{K}} \boldsymbol{s}_{i+1} \\ \boldsymbol{0} \end{pmatrix} \\
&= \begin{pmatrix} \boldsymbol{d}_{i+1} \\ \boldsymbol{0} \end{pmatrix}
\end{aligned}$$

where we used that $\tilde{\eta}_j = \tilde{\boldsymbol{s}}_j^\intercal(k(\tilde{\boldsymbol{X}}, \tilde{\boldsymbol{X}}) + \sigma^2 \boldsymbol{I}_{n+n'})\tilde{\boldsymbol{d}}_j = \boldsymbol{s}_j^\intercal \hat{\boldsymbol{K}}\boldsymbol{d}_j = \eta_j$. This proves eq. (S51). Now recognize that

$$\begin{aligned}
\tilde{\alpha}_j &= \tilde{\boldsymbol{s}}_j^\intercal \tilde{\boldsymbol{r}}_j = \tilde{\boldsymbol{s}}_j^\intercal(\tilde{\boldsymbol{y}} - \tilde{\boldsymbol{\mu}} - \tilde{\boldsymbol{K}}\tilde{\boldsymbol{C}}_i(\tilde{\boldsymbol{y}} - \tilde{\boldsymbol{\mu}})) \\
&= \tilde{\boldsymbol{s}}_j^\intercal(\tilde{\boldsymbol{y}} - \tilde{\boldsymbol{\mu}} - (\tilde{\boldsymbol{K}} + \sigma^2 \boldsymbol{I})\sum_{\ell=1}^{j}\frac{1}{\tilde{\eta}_\ell}\tilde{\boldsymbol{d}}_\ell \tilde{\boldsymbol{d}}_\ell^\intercal(\tilde{\boldsymbol{y}} - \tilde{\boldsymbol{\mu}})) \\
&= \boldsymbol{s}_j^\intercal(\boldsymbol{y} - \boldsymbol{\mu}) - \sum_{\ell=1}^{j}\frac{1}{\eta_\ell}\boldsymbol{s}_j^\intercal \hat{\boldsymbol{K}}\boldsymbol{d}_\ell \boldsymbol{d}_\ell^\intercal(\boldsymbol{y} - \boldsymbol{\mu}) \\
&= \boldsymbol{s}_j^\intercal(\boldsymbol{y} - \boldsymbol{\mu} - \hat{\boldsymbol{K}}\boldsymbol{C}_j(\boldsymbol{y} - \boldsymbol{\mu})) \\
&= \boldsymbol{s}_j^\intercal \boldsymbol{r}_j \\
&= \alpha_j
\end{aligned}$$

Therefore, we finally have that

$$\begin{aligned}
\tilde{\mu}_i(\cdot) &= \mu(\cdot) + k(\cdot, \tilde{\boldsymbol{X}})\tilde{\boldsymbol{v}}_i = \mu(\cdot) + k(\cdot, \tilde{\boldsymbol{X}})\sum_{j=1}^{i}\frac{\tilde{\alpha}_j}{\tilde{\eta}_j}\tilde{\boldsymbol{d}}_j \\
&= \mu(\cdot) + k(\cdot, \boldsymbol{X})\boldsymbol{v}_i
\end{aligned}$$

as well as

$$\begin{aligned}
\tilde{k}_i(\cdot, \cdot) &= k(\cdot, \cdot) - k(\cdot, \tilde{\boldsymbol{X}})\tilde{\boldsymbol{C}}_i k(\tilde{\boldsymbol{X}}, \cdot) = k(\cdot, \cdot) - k(\cdot, \tilde{\boldsymbol{X}})\sum_{j=1}^{i}\frac{1}{\tilde{\eta}_j}\tilde{\boldsymbol{d}}_j(\tilde{\boldsymbol{d}}_j)^\intercal k(\tilde{\boldsymbol{X}}, \cdot) \\
&= k(\cdot, \cdot) - k(\cdot, \boldsymbol{X})\sum_{j=1}^{i}\frac{1}{\eta_j}\boldsymbol{d}_j(\boldsymbol{d}_j)^\intercal k(\boldsymbol{X}, \cdot) = k(\cdot, \cdot) - k(\cdot, \boldsymbol{X})\boldsymbol{C}_i k(\boldsymbol{X}, \cdot) = k_i(\cdot, \cdot).
\end{aligned}$$

$\square$

**Remark S2** (Streaming Gaussian Processes)
Theorem S7 shows that any variant of IterGP can be used in the online setting where data arrives sequentially *while* the algorithm is running. Now, if we assume data points arrive one at a time, we choose unit vector actions (IterGP-Chol) and perform one iteration of Algorithm 1 after each data point, then Algorithm 1 simply computes the mathematical GP posterior.

## S2.2 Approximation of Representer Weights

**Proposition 2** (Relative Error Bound for the Representer Weights)
*For any choice of actions a relative error bound $\rho(i)$, s.t. $\|\boldsymbol{v}_* - \boldsymbol{v}_i\|_{\hat{\boldsymbol{K}}} \leq \rho(i)\|\boldsymbol{v}_*\|_{\hat{\boldsymbol{K}}}$ is given by*

$$\rho(i) = (\bar{\boldsymbol{v}}_*^\intercal \underbrace{(\boldsymbol{I} - \boldsymbol{C}_i \hat{\boldsymbol{K}})}_{\text{projection onto } \text{span}\{\boldsymbol{S}_i\}^{\perp \hat{\boldsymbol{K}}}} \bar{\boldsymbol{v}}_*)^{\frac{1}{2}} \leq \lambda_{\max}(\boldsymbol{I} - \boldsymbol{C}_i \hat{\boldsymbol{K}}) \leq 1 \tag{9}$$

*where $\bar{\boldsymbol{v}}_* = \boldsymbol{v}_*/\|\boldsymbol{v}_*\|_{\hat{\boldsymbol{K}}}$. If the actions $\{\boldsymbol{s}_i\}_{i=1}^{n}$ are linearly independent, then $\rho(i) \leq \delta_{n=i}$.*

*Proof.* Define $\boldsymbol{H}_i = \boldsymbol{\Sigma}_i \hat{\boldsymbol{K}} = \boldsymbol{I} - \boldsymbol{C}_i \hat{\boldsymbol{K}}$. We have by Lemma S2, that

$$\|\boldsymbol{v}_* - \boldsymbol{v}_i\|_{\hat{\boldsymbol{K}}}^2 = \|\boldsymbol{H}_i \boldsymbol{v}_*\|_{\hat{\boldsymbol{K}}}^2 = (\boldsymbol{H}_i \boldsymbol{v}_*)^\intercal \hat{\boldsymbol{K}}\boldsymbol{H}_i \boldsymbol{v}_* \stackrel{(\text{S44})}{=} \boldsymbol{v}_*^\intercal \boldsymbol{H}_i \boldsymbol{v}_* = \bar{\boldsymbol{v}}_*^\intercal \boldsymbol{H}_i \bar{\boldsymbol{v}}_* \|\boldsymbol{v}_*\|_{\hat{\boldsymbol{K}}}^2$$

This proves the first equality of Proposition 2. Further it holds that

$$\begin{aligned}
\|\boldsymbol{H}_i \boldsymbol{v}_*\|_{\hat{\boldsymbol{K}}} &= \|\hat{\boldsymbol{K}}^{\frac{1}{2}}\boldsymbol{H}_i \boldsymbol{v}_*\|_2 = \|(\boldsymbol{I} - \hat{\boldsymbol{K}}^{\frac{1}{2}}\boldsymbol{C}_i \hat{\boldsymbol{K}}^{\frac{1}{2}})\hat{\boldsymbol{K}}^{\frac{1}{2}}\boldsymbol{v}_*\|_2 \leq \|\boldsymbol{I} - \hat{\boldsymbol{K}}^{\frac{1}{2}}\boldsymbol{C}_i \hat{\boldsymbol{K}}^{\frac{1}{2}}\|_2 \|\boldsymbol{v}_*\|_{\hat{\boldsymbol{K}}} \\
&= \lambda_{\max}(\boldsymbol{I} - \hat{\boldsymbol{K}}^{\frac{1}{2}}\boldsymbol{C}_i \hat{\boldsymbol{K}}^{\frac{1}{2}})\|\boldsymbol{v}_*\|_{\hat{\boldsymbol{K}}}.
\end{aligned}$$

Now by Weyl's inequality and the fact that $\hat{\boldsymbol{K}}^{\frac{1}{2}}\boldsymbol{C}_i \hat{\boldsymbol{K}}^{\frac{1}{2}}$ is positive semi-definite, it holds that

$$\lambda_{\max}(\boldsymbol{H}_i) = \lambda_{\max}(\boldsymbol{I} - \hat{\boldsymbol{K}}^{\frac{1}{2}}\boldsymbol{C}_i \hat{\boldsymbol{K}}^{\frac{1}{2}}) \leq \lambda_{\max}(\boldsymbol{I}) - \lambda_{\min}(\hat{\boldsymbol{K}}^{\frac{1}{2}}\boldsymbol{C}_i \hat{\boldsymbol{K}}^{\frac{1}{2}}) \leq 1.$$

Now, recall that similar matrices $\boldsymbol{A}$ and $\boldsymbol{B} = \boldsymbol{P}^{-1}\boldsymbol{A}\boldsymbol{P}$ have the same eigenvalues. Therefore

$$\boldsymbol{I} - \hat{\boldsymbol{K}}^{\frac{1}{2}}\boldsymbol{C}_i\hat{\boldsymbol{K}}^{\frac{1}{2}} = \hat{\boldsymbol{K}}^{\frac{1}{2}}(\boldsymbol{I} - \boldsymbol{C}_i\hat{\boldsymbol{K}})\hat{\boldsymbol{K}}^{-\frac{1}{2}}$$

and $\boldsymbol{I} - \boldsymbol{C}_i\hat{\boldsymbol{K}}$ have the same eigenvalues. Finally, since by eq. (S39) $\boldsymbol{H}_i$ is a projection onto $\mathrm{span}\{\boldsymbol{S}_i\}^{\perp_{\hat{K}}}$, it has full rank at iteration $n$ if the actions are linearly independent and therefore $\lambda_{\max}(\boldsymbol{H}_n) = 1$. This proves the claim. $\qquad\square$

## S2.3 Convergence Analysis of the Posterior Mean Approximation

**Theorem 1** (Convergence in RKHS Norm of the Posterior Mean Approximation)
*Let $\mathcal{H}_k$ be the RKHS associated with kernel $k(\cdot,\cdot)$, $\sigma^2 > 0$ and let $\mu_* - \mu \in \mathcal{H}_k$ be the unique solution to the regularized empirical risk minimization problem*

$$\arg\min_{f\in\mathcal{H}_k} \frac{1}{n}\Big(\sum_{j=1}^{n}(f(\boldsymbol{x}_j) - y_j + \mu(\boldsymbol{x}_j))^2 + \sigma^2\|f\|_{\mathcal{H}_k}^2\Big) \tag{11}$$

*which is equivalent to the mathematical posterior mean up to shift by the prior $\mu$ [e.g. 1, Sec. 6.2]. Then for $i \in \{0,\dots,n\}$ the posterior mean $\mu_i(\cdot)$ computed by Algorithm 1 satisfies*

$$\boxed{\|\mu_* - \mu_i\|_{\mathcal{H}_k} \le \rho(i)c(\sigma^2)\|\mu_* - \mu_0\|_{\mathcal{H}_k}} \tag{12}$$

*where $\mu_0 = \mu$ is the prior mean and the constant $c(\sigma^2) = \sqrt{1 + \frac{\sigma^2}{\lambda_{\min}(\boldsymbol{K})}} \to 1$ as $\sigma^2 \to 0$.*

*Proof.* Let $\rho(i)$ such that $\|\boldsymbol{v}_* - \boldsymbol{v}_i\|_{\hat{K}} \le \rho(i)\|\boldsymbol{v}_* - \boldsymbol{v}_0\|_{\hat{K}}$, where $\boldsymbol{v}_0 = \boldsymbol{0}$. Then, we have for $i \in \{0,\dots,n\}$, that

$$\|\boldsymbol{v}_* - \boldsymbol{v}_i\|_{\boldsymbol{K}}^2 \le \|\boldsymbol{v}_* - \boldsymbol{v}_i\|_{\hat{K}}^2 \le \rho(i)^2\|\boldsymbol{v}_* - \boldsymbol{v}_0\|_{\hat{K}}^2$$

$$= \rho(i)^2\Big(\|\boldsymbol{v}_* - \boldsymbol{v}_0\|_{\boldsymbol{K}}^2 + \sigma^2\frac{1}{\lambda_{\min}(\boldsymbol{K})}\underbrace{\lambda_{\min}(\boldsymbol{K})\|\boldsymbol{v}_* - \boldsymbol{v}_0\|_2^2}_{\le\|\boldsymbol{v}_* - \boldsymbol{v}_0\|_{\boldsymbol{K}}^2}\Big)$$

$$\le \rho(i)^2\Big(1 + \frac{\sigma^2}{\lambda_{\min}(\boldsymbol{K})}\Big)\|\boldsymbol{v}_* - \boldsymbol{v}_0\|_{\boldsymbol{K}}^2$$

Now by assumption $\mu_i(\cdot) = \mu(\cdot) + \sum_{j=1}^{n}(\boldsymbol{v}_i)_j k(\cdot,\boldsymbol{x}_j) = \mu(\cdot) + k(\cdot,\boldsymbol{X})\boldsymbol{C}_i\boldsymbol{y}$. By the reproducing property we obtain for $\Delta = \boldsymbol{v}_* - \boldsymbol{v}_i$ that

$$\|\boldsymbol{v}_* - \boldsymbol{v}_i\|_{\boldsymbol{K}}^2 = \Delta^{\intercal}\boldsymbol{K}\Delta$$

$$= \sum_{\ell=1}^{n}\sum_{j=1}^{n}\Delta_\ell\Delta_j k(\boldsymbol{x}_\ell,\boldsymbol{x}_j)$$

$$= \sum_{\ell=1}^{n}\sum_{j=1}^{n}\Delta_\ell\Delta_j\langle k(\cdot,\boldsymbol{x}_\ell), k(\cdot,\boldsymbol{x}_j)\rangle_{\mathcal{H}_k} \qquad\text{$k$ is the reproducing kernel of $\mathcal{H}_k$}$$

$$= \langle \sum_{\ell=1}^{n}\Delta_\ell k(\cdot,\boldsymbol{x}_\ell), \sum_{j=1}^{n}\Delta_j k(\cdot,\boldsymbol{x}_j)\rangle_{\mathcal{H}_k}$$

$$= \Big\|\sum_{\ell=1}^{n}\Delta_\ell k(\cdot,\boldsymbol{x}_\ell)\Big\|_{\mathcal{H}_k}^2$$

$$= \Big\|\sum_{\ell=1}^{n}(\boldsymbol{v}_*)_\ell k(\cdot,\boldsymbol{x}_\ell) - \sum_{\ell=1}^{n}(\boldsymbol{v}_i)_\ell k(\cdot,\boldsymbol{x}_\ell)\Big\|_{\mathcal{H}_k}^2$$

$$= \|\mu_* - \mu_i\|_{\mathcal{H}_k}^2 \qquad\text{See Theorem 3.4 in Kanagawa et al. [36]}$$

Combining the above and setting $c(\sigma^2) = 1 + \frac{\sigma^2}{\lambda_{\min}(\boldsymbol{K})}$ we obtain

$$\|\mu_* - \mu_i\|_{\mathcal{H}_k} = \|\boldsymbol{v}_* - \boldsymbol{v}_i\|_{\boldsymbol{K}} \le \rho(i)c(\sigma^2)\|\boldsymbol{v}_* - \boldsymbol{v}_0\|_{\boldsymbol{K}} = \rho(i)c(\sigma^2)\|\mu_* - \mu_0\|_{\mathcal{H}_k}.$$

$$\square$$

## S2.4 Combined Uncertainty as Worst Case Error

**Theorem 2** (Combined and Computational Uncertainty as Worst Case Errors)
*Let $\sigma^2 \geq 0$ and let $k_i(\cdot, \cdot) = k_*(\cdot, \cdot) + k_i^{\text{comp}}(\cdot, \cdot)$ be the combined uncertainty computed by Algorithm 1. Then, for any $\boldsymbol{x} \in \mathcal{X}$ (assuming $\boldsymbol{x} \notin \boldsymbol{X}$ if $\sigma^2 > 0$) we have*

$$\sup_{g \in \mathcal{H}_{k^\sigma}: \|g\|_{\mathcal{H}_{k^\sigma}} \leq 1} \underbrace{\overbrace{g(\boldsymbol{x}) - \mu_*^g(\boldsymbol{x})}^{\text{error of approximate posterior mean}} + \underbrace{\mu_*^g(\boldsymbol{x}) - \mu_i^g(\boldsymbol{x})}_{\text{computational error}}}_{\text{error of math. post. mean}} = \sqrt{k_i(\boldsymbol{x}, \boldsymbol{x}) + \sigma^2}, \quad \text{and} \qquad (13)$$

$$\sup_{g \in \mathcal{H}_{k^\sigma}: \|g\|_{\mathcal{H}_{k^\sigma}} \leq 1} \underbrace{\mu_*^g(\boldsymbol{x}) - \mu_i^g(\boldsymbol{x})}_{\text{computational error}} = \sqrt{k_i^{\text{comp}}(\boldsymbol{x}, \boldsymbol{x})} \qquad (14)$$

*where $\mu_*^g(\cdot) = k(\cdot, \boldsymbol{X})\hat{\boldsymbol{K}}^{-1}g(\boldsymbol{X})$ is the mathematical and $\mu_i^g(\cdot) = k(\cdot, \boldsymbol{X})\boldsymbol{C}_i g(\boldsymbol{X})$ IterGP's posterior mean for the latent function $g \in \mathcal{H}_{k^\sigma}$. If $\sigma^2 = 0$, then the above also holds for $\boldsymbol{x} \in \boldsymbol{X}$.*

*Proof.* Let $\boldsymbol{x}_0 = \boldsymbol{x}$, $c_0 = 1$ and $c_j = -(\boldsymbol{C}_i k^\sigma(\boldsymbol{X}, \boldsymbol{x}))_j$ for $j = 1, \ldots n$, where $k^\sigma(\cdot, \cdot) := k(\cdot, \cdot) + \sigma^2\delta(\cdot, \cdot)$. Then by Lemma 3.9 of Kanagawa et al. [36], it holds that

$$\left( \sup_{g \in \mathcal{H}_{k^\sigma}: \|g\|_{\mathcal{H}_{k^\sigma}} \leq 1} (g(\boldsymbol{x}) - \mu_i^g(\boldsymbol{x})) \right)^2 = \left( \sup_{g \in \mathcal{H}_{k^\sigma}: \|g\|_{\mathcal{H}_{k^\sigma}} \leq 1} \sum_{j=0}^n c_j g(\boldsymbol{x}_j) \right)^2$$

$$= \left\| k^\sigma(\cdot, \boldsymbol{x}_0) - \sum_{j=1}^n k(\boldsymbol{x}, \boldsymbol{x}_j)\boldsymbol{C}_i k^\sigma(\cdot, \boldsymbol{x}_j) \right\|_{\mathcal{H}_{k^\sigma}}^2$$

$$= \| k^\sigma(\cdot, \boldsymbol{x}) - k(\boldsymbol{x}, \boldsymbol{X})\boldsymbol{C}_i k^\sigma(\boldsymbol{X}, \cdot) \|_{\mathcal{H}_{k^\sigma}}^2$$

$$= \langle k^\sigma(\cdot, \boldsymbol{x}), k^\sigma(\cdot, \boldsymbol{x}) \rangle_{\mathcal{H}_{k^\sigma}} - 2\langle k^\sigma(\cdot, \boldsymbol{x}), k(\boldsymbol{x}, \boldsymbol{X})\boldsymbol{C}_i k^\sigma(\boldsymbol{X}, \cdot) \rangle_{\mathcal{H}_{k^\sigma}}$$
$$+ \langle k(\boldsymbol{x}, \boldsymbol{X})\boldsymbol{C}_i k^\sigma(\boldsymbol{X}, \cdot), k(\boldsymbol{x}, \boldsymbol{X})\boldsymbol{C}_i k^\sigma(\boldsymbol{X}, \cdot) \rangle_{\mathcal{H}_{k^\sigma}}$$

Now by the reproducing property, it follows that

$$= k^\sigma(\boldsymbol{x}, \boldsymbol{x}) - 2k^\sigma(\boldsymbol{x}, \boldsymbol{X})\boldsymbol{C}_i k^\sigma(\boldsymbol{X}, \boldsymbol{x}) + k^\sigma(\boldsymbol{x}, \boldsymbol{X})\boldsymbol{C}_i k^\sigma(\boldsymbol{X}, \boldsymbol{X})\boldsymbol{C}_i k^\sigma(\boldsymbol{X}, \boldsymbol{x})$$

If $\sigma^2 > 0$ and $\boldsymbol{x} \neq \boldsymbol{x}_j$ or if $\sigma^2 = 0$, it holds that $k^\sigma(\boldsymbol{x}, \boldsymbol{X}) = k(\boldsymbol{x}, \boldsymbol{X})$. Further by definition $k^\sigma(\boldsymbol{X}, \boldsymbol{X}) = \hat{\boldsymbol{K}}$ and finally by (S42), it holds that $\boldsymbol{C}_i \hat{\boldsymbol{K}}\boldsymbol{C}_i = \boldsymbol{C}_i$. Therefore we have

$$= k(\boldsymbol{x}, \boldsymbol{x}) + \sigma^2 - 2k(\boldsymbol{x}, \boldsymbol{X})\boldsymbol{C}_i k(\boldsymbol{X}, \boldsymbol{x}) + k(\boldsymbol{x}, \boldsymbol{X})\boldsymbol{C}_i \hat{\boldsymbol{K}}\boldsymbol{C}_i k(\boldsymbol{X}, \boldsymbol{x})$$
$$= k(\boldsymbol{x}, \boldsymbol{x}) - k(\boldsymbol{x}, \boldsymbol{X})\boldsymbol{C}_i k(\boldsymbol{X}, \boldsymbol{x}) + \sigma^2$$
$$= k_i(\boldsymbol{x}, \boldsymbol{x}) + \sigma^2$$

We prove eq. (14) by an analogous argument. Choose $c_j := ((\hat{\boldsymbol{K}}^{-1} - \boldsymbol{C}_i)k^\sigma(\boldsymbol{X}, \boldsymbol{x}))_j$. We have

$$\left( \sup_{g \in \mathcal{H}_{k^\sigma}: \|g\|_{\mathcal{H}_{k^\sigma}} \leq 1} (\mu_*^g(\boldsymbol{x}) - \mu_i^g(\boldsymbol{x})) \right)^2 = \left( \sup_{g \in \mathcal{H}_{k^\sigma}: \|g\|_{\mathcal{H}_{k^\sigma}} \leq 1} \sum_{j=0}^n c_j g(\boldsymbol{x}_j) \right)^2$$

$$= \left\| \sum_{j=1}^n k(\boldsymbol{x}, \boldsymbol{x}_j)(\hat{\boldsymbol{K}}^{-1} - \boldsymbol{C}_i)k^\sigma(\cdot, \boldsymbol{x}_j) \right\|_{\mathcal{H}_{k^\sigma}}^2$$

$$= \| k(\boldsymbol{x}, \boldsymbol{X})(\hat{\boldsymbol{K}}^{-1} - \boldsymbol{C}_i)k^\sigma(\boldsymbol{X}, \cdot) \|_{\mathcal{H}_{k^\sigma}}^2$$

$$= k^\sigma(\boldsymbol{x}, \boldsymbol{X})\hat{\boldsymbol{K}}^{-1}\hat{\boldsymbol{K}}\hat{\boldsymbol{K}}^{-1}k^\sigma(\boldsymbol{X}, \boldsymbol{x}) - 2k^\sigma(\boldsymbol{x}, \boldsymbol{X})\hat{\boldsymbol{K}}^{-1}\hat{\boldsymbol{K}}\boldsymbol{C}_i k^\sigma(\boldsymbol{X}, \boldsymbol{x}) + k^\sigma(\boldsymbol{x}, \boldsymbol{X})\boldsymbol{C}_i \hat{\boldsymbol{K}}\boldsymbol{C}_i k^\sigma(\boldsymbol{X}, \boldsymbol{x})$$

Again, we use that $k^\sigma(\boldsymbol{x}, \boldsymbol{X}) = k(\boldsymbol{x}, \boldsymbol{X})$ by assumption and (S42). Therefore

$$= k(\boldsymbol{x}, \boldsymbol{X})(\hat{\boldsymbol{K}}^{-1} - \boldsymbol{C}_i)k(\boldsymbol{X}, \boldsymbol{x})$$
$$= k_i^{\text{comp}}(\boldsymbol{x}, \boldsymbol{x})$$

This concludes the proof. $\qquad \square$

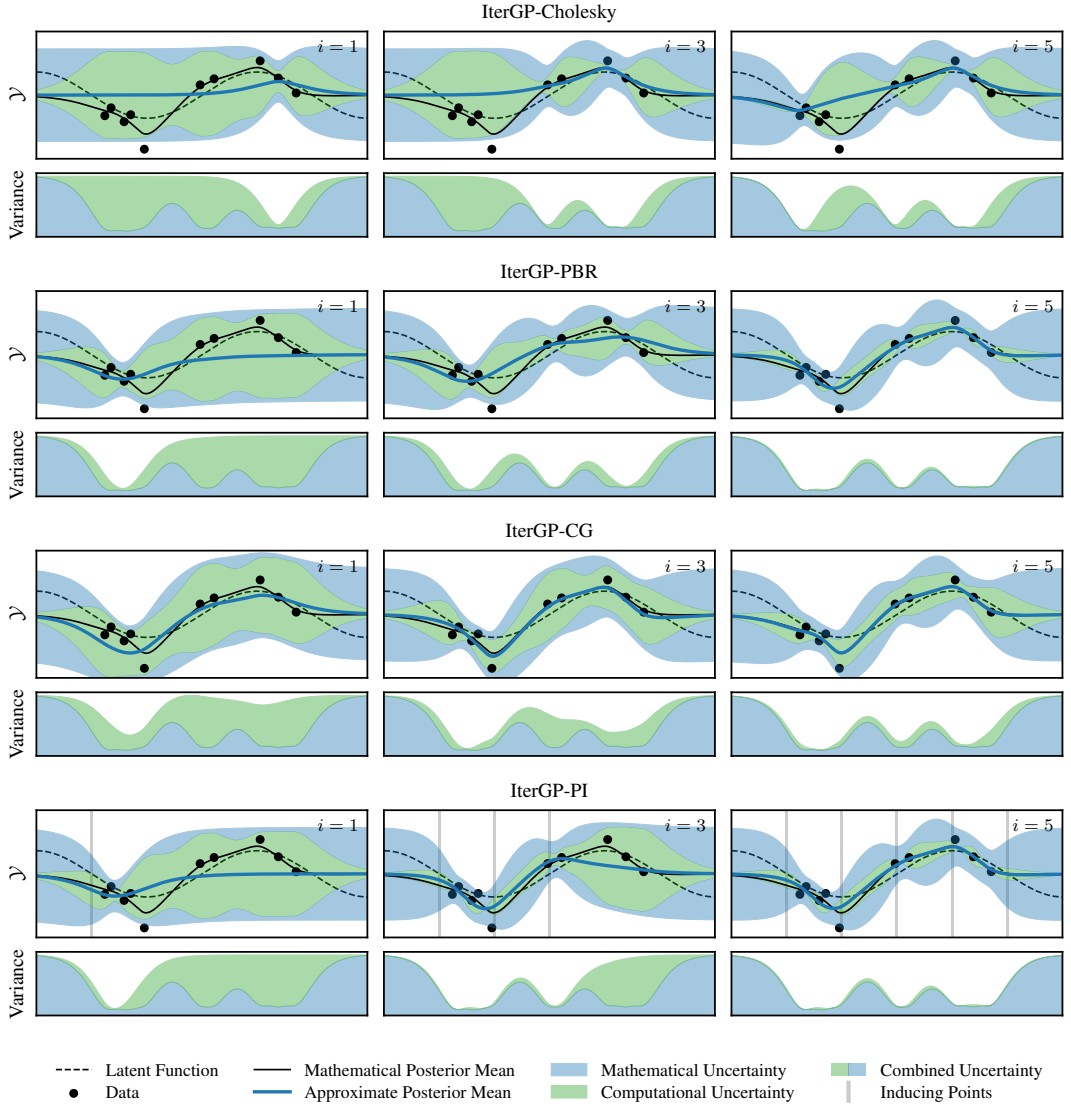

Figure S3: *Illustration of IterGP analogs of commonly used GP approximations.*

## S3 Implementation of Algorithm 1

### S3.1 Policy Choice

As illustrated in Figure 2, the choice of policy of Algorithm 1 determines where computation in input space is targeted and therefore where the combined posterior contracts first. However, the policy also determines whether the error in the posterior mean or (co-)variance are predominantly reduced first, as Figure S3 shows (cf. IterGP-Chol and IterGP-PBR). Therefore the policy choice is application-dependent. If I am primarily interested in the predictive mean, I may select residual actions (IterGP-CG). If downstream I am making use of the predictive uncertainty, I may want to contract uncertainty globally as quickly as possible at the expense of predictive accuracy (IterGP-PI). Such a choice is not unique to IterGP, but necessary whenever we select a GP approximation. What IterGP adds is computation-aware, meaningful uncertainty quantification in the sense of Corollary 1 no matter the choice of policy.

## S3.2 Stopping Criterion

In our implementation of Algorithm 1 we use the following two stopping criteria. Our computational budget can be directly controlled by specifying a *maximum number of iterations*, since each iteration of IterGP needs the same number of matrix-vector multiplies. Alternatively, we terminate if the *absolute or relative norm of the residual* are sufficiently small, i.e. if

$$\|\boldsymbol{r}_i\|_2 < \delta_{\text{abstol}} \qquad \text{or} \qquad \|\boldsymbol{r}_i\|_2 < \delta_{\text{reltol}}\|\boldsymbol{y}\|_2. \tag{S53}$$

Of course other choices are possible. From a probabilistic numerics standpoint one may want to terminate once the combined marginal uncertainty at the training data is sufficiently small relative to the observation noise.

## S3.3 Efficient Sampling from the Combined Posterior

Sampling from an exact GP posterior has cubic cost $\mathcal{O}(n_\diamond^3)$ in the number of evaluation points $n_\diamond$, which is prohibitive for many useful downstream applications such as numerical integration over the posterior using Monte-Carlo methods. Wilson et al. [46, 47] recently showed how to make use of *Matheron's rule* [45, 66, 67] to efficiently sample from a GP posterior by sampling from the prior and then performing a pathwise update. We can directly make use of this strategy since Algorithm 1 computes a low-rank approximation to the precision matrix. Assume we are given a draw $f'_{\text{prior}} \in \mathcal{H}_k^\theta$ from the prior[3] such that $\boldsymbol{y}' \sim \mathcal{N}(f'_{\text{prior}}(\boldsymbol{X}), \sigma^2 \boldsymbol{I})$ constitutes a draw from the prior predictive. Then

$$f'(\cdot) = f'_{\text{prior}}(\cdot) + k(\cdot, \boldsymbol{X})\boldsymbol{C}_i(\boldsymbol{y} - \boldsymbol{y}') \tag{S54}$$

is a draw from the combined posterior by Matheron's rule, which we can evaluate in $\mathcal{O}(n_\diamond n i)$ for $n_\diamond$ evaluation points, since $\boldsymbol{C}_i$ has rank $i$.

## S4 Additional Experimental Results

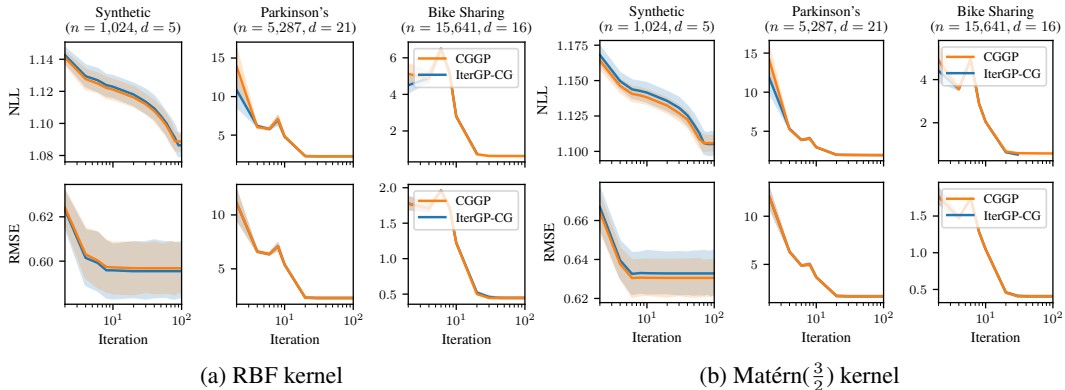

Figure S4: *Generalization of CGGP and its closest IterGP analog.* GP regression using an RBF and Matérn($\frac{3}{2}$) kernel on UCI datasets. The plot shows the average generalization error in terms of NLL and RMSE for an increasing number of solver iterations. The posterior mean of IterGP-CG and CGGP is identical, which explains the identical RMSE.

---

[3]In infinite dimensional reproducing kernel Hilbert spaces samples $f \sim \mathcal{GP}(\mu, k)$ from a Gaussian process almost surely do not lie in the RKHS $\mathcal{H}_k$ [Cor. 4.10, 36]. However, there exists $f' \in \mathcal{H}_k^\theta$ in a larger RKHS $\mathcal{H}_k^\theta \supset \mathcal{H}_k$ such that $f'(\boldsymbol{x}) = f(\boldsymbol{x})$ with probability 1 [Thm. 4.12, 36].

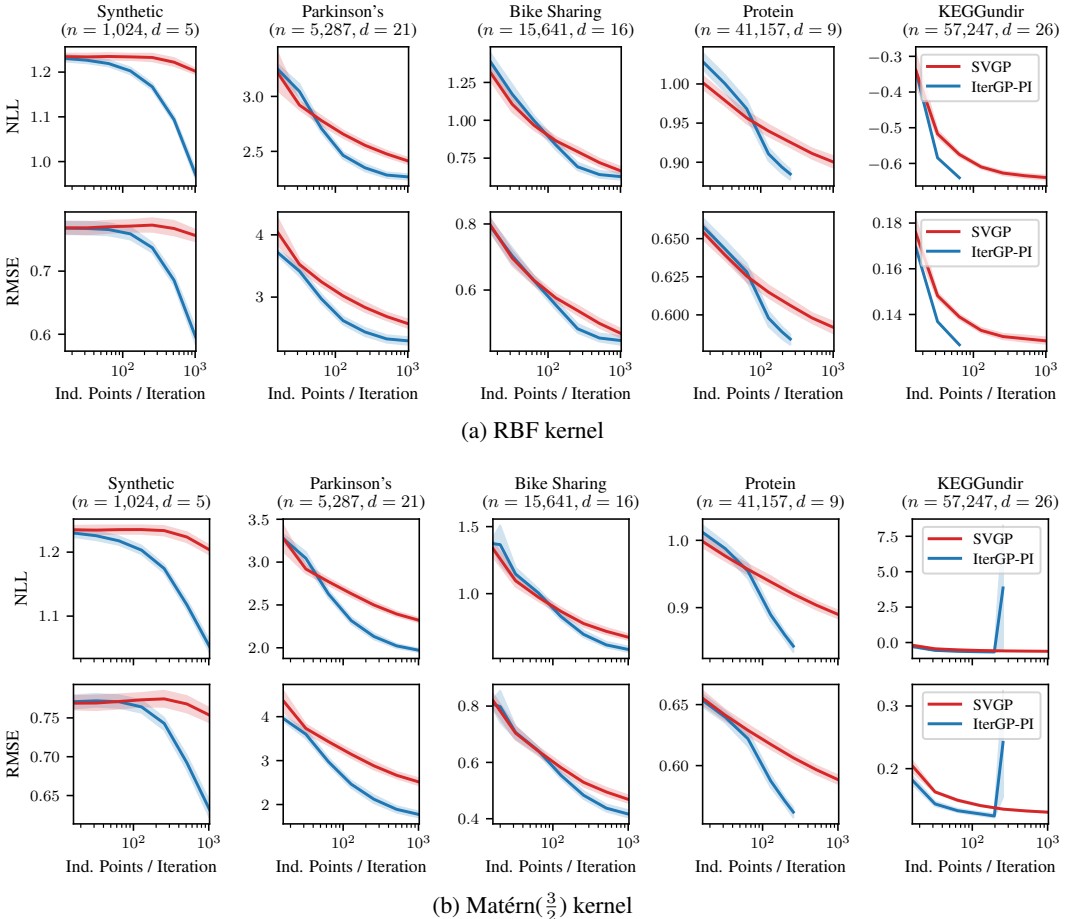

Figure S5: *Generalization of SVGP and its closest IterGP analog.* GP regression using an RBF and Matérn($\frac{3}{2}$) kernel on UCI datasets. The plot shows the average generalization error in terms of NLL and RMSE for an increasing number of identical inducing points. After a small number of inducing points relative to the size of the training data, IterGP has significantly lower generalization error than SVGP. For the "KEGGundir" dataset after $\approx 128$ iterations we observe numerical instability in some runs when computing the combined posterior of IterGP using a Matérn($\frac{3}{2}$) kernel.