# OpenReview forum: "Posterior and Computational Uncertainty in Gaussian Processes"
_NeurIPS.cc/2022/Conference — NeurIPS 2022 Accept_

### Official Review · Reviewer_pTot · 2022-07-08

**Rating:** 6
**Confidence:** 4
**Soundness:** 2 fair
**Presentation:** 3 good
**Contribution:** 2 fair

**Summary:**

The paper proposes an approximate inference technique, called IterGP, for Gaussian progress (GP) regression. Methodologically, the key algorithm (Algorithm 1) provably constructs increasingly more accurate approximations (of the GP posterior) as the runtime increases. Each approximation also comes with a notion of approximation error (in the sense of Theorem 2). Empirically, IterGP can be computationally cheaper, but just as accurate as existing methods (Figure 4), or more expensive but more accurate compared to existing methods (Figure 5).

**Questions:**

I would like the authors to address my questions regarding the “quality” subsection of the “Strengths and Weaknesses” section. My biggest concern is being confused about the relationship between combined uncertainty and the approximate posterior variance. The answer to this question is important, because if the combined uncertainty is the same as the approximate posterior variance, then the claim in the abstract “[…] due to limited computation, is entirely ignored when using the approximate posterior” is not accurate, since approximate GP automatically accounts for both computational and mathematical uncertainties, through the approximate posterior variance / combined uncertainty. More generally, I don't see the benefit of the so-called combined uncertainty, it it's just equal to the approximate posterior variance. My confusion also spawned my question about whether Theorem 2 applies to other approximate GP techniques. It also makes me question whether the claim in Figure 1 is valid: if it were true that Theorem 2 also applies to SVGP, then the message of modelling computational uncertainty improves GP approximation is incorrect. Perhaps the cause of IterGP-PI performing better than SVGP is due to the later using inducing points, which can be inaccurate.

The list of typos I’ve found are as follows
- Lines 83-84: Equation 2 is true if the prior mean is zero.

- Lines 96:97: the contraction only happens for a well chosen set of policies $s_i$, not in general

- Lines 119-120: it’s not true that the algorithm is online. Line 9 of Algorithm 1 requires the whole kernel matrix $\hat{K}$.


**Limitations:**

I do not see any negative societal implications to this line of work.

**Strengths And Weaknesses:**

I think the paper’s key ideas are original and strong, but I have some concerns about the quality and clarity of the manuscript.

# Originality
Section 2 is a novel way of using probabilistic linear solvers (PLS) to perform GP regression. Section 2 treats the so called representer weights as unknown/random, whereas existing works treat either the kernel matrix or the inverse of the kernel matrix as random. The randomness here is due to limited computation. Since the posterior mean at a test point is a function of the representer weight (Equation 2), the randomness in the representer weight propagates into the posterior mean and is computable as in Equation 6. The update equations 4 and 5 give the general skeleton of how to reduce the randomness in the representer weights.

# Quality
In terms of strengths, the visualizations and plots in the paper are very legible and convey the message well. Figure 1 panel b breaks down uncertainty into two constituent parts. Figure 4 demonstrate how IterGP is cheaper than conjugate gradient (CG) GP but just as accurate. Similarly, Figure 5 is very easy to read and understand the takeaway.

In terms of room for improvement, I think the paper is missing three kinds of discussions.
- The first is the impact of the policy. Although Table 1 mentions a list of policies, the rest of the paper does not discuss the pros and cons of different policies, or give recommendations on which to use. I do understand that in a sense the actions s_i are hyper-parameters of Algorithm 1, and tuning hyper-parameters is potentially its own problem.
- The second is distinguishing the combined uncertainty (in Equation 6) from the regular approximate posterior variance at test points. What I mean by approximate posterior variance is that, while the true posterior variance at $x_*$ is
$$k(x_*, x_*) – k(x_*, X) \hat{K}^{-1} k(X, x_*),$$
When we have an approximation of the inverse of the kernel matrix, say, $L \approx \hat{K}^{-1}$, then we announce the approximate posterior variance to be
$$k(x_*, x_*) – k(x_*, X) L k(X, x_*),$$
My current impression is that the combined uncertainty is exactly the approximate posterior variance for the choice $L = C_i$, where $C_i$ is constructed using the IterGP iterative procedure. But more generally, all approximate GP approaches that have approximations to $\hat{K}^{-1}$ will also have the combined uncertainty, and it’s equal to the approximate posterior variance that they report.
- The final discussion is whether Theorem 2, in particular Equation 13, applies to other kinds of approximate GP inference techniques. Skimming through Theorem 2’s proof, there is no part of the proof that relies on the actual structure of iterGP, in the sense that the statement will hold if we replace $C_i$ with any other approximation of $\hat{K}^{-1}$. In that case, since other approximate GP inference techniques have their own approximation of $\hat{K}^{-1}$, their approximate posterior variance can also be used to give worst-case bounds in the sense of Theorem 2.

# Clarity
The paper is well-written. There are some negligible typos that I will highlight in the “Questions” section.

# Significance
Advances in scalable and accurate GP inference are important since GP regression is a staple of modern data analysis.

---

> ### Author Response · Authors · 2022-08-02
> **Author Response to Reviewer pTot**
>
>
> Thank you for your review and the discussion points you raise. We want to first point out a key technical oversight in your review that may have lead to a more negative opinion of this work: **Theorem 2 does not apply to other GP approximations, which therefore do not quantify combined and computational uncertainty.** See our detailed responses below. If we answer your questions, we would appreciate if you'd consider updating your score. If anything remains unclear, we are happy to clarify in the discussion period.
>
> ## Detailed Response
>
> ### Q1: Impact of the Policy
> > The first [missing discussion] is the impact of the policy.
>
> As is illustrated in Figure S2 the choice of policy determines
>
> - where computation in input space is targeted, and thus
> - where the combined posterior contracts first (see e.g. IterGP-Chol), and
> - whether the error in the posterior mean or (co-)variance are predominantly reduced first (compare IterGP-CG vs IterGP-PBR / IterGP-PI)
>
> Thus the policy choice is application-dependent. If I am solely interested in the predictive mean, I may choose IterGP-CG. If my goal is UQ (e.g. for active learning) I may choose IterGP-PI. Such a choice is not unique to IterGP, but necessary whenever we select a GP approximation. What IterGP adds is computation-aware, meaningful uncertainty quantification (in the sense of Theorem 2).
>
> ### Q2: Combined Uncertainty vs. Approximate Posterior Variance
>
> > I don't see the benefit of the so-called combined uncertainty, if it's just equal to the approximate posterior variance.
>
> **The combined uncertainty** (with $C_i \approx \hat{K}^{-1}$) **as opposed to the approximate posterior variance** (arbitrary $L \approx \hat{K}^{-1}$) **guarantees that**
>
> 1. the marginal variance of the exact GP is never underestimated (see eqn. 6 and Figure 1 bottom).
> 2. the combined uncertainty is a worst case bound on the error to the true latent function (see eqn. 13)
>
> In that sense, the combined uncertainty describes precisely how uncertain we should be, given our approximate posterior mean (see Theorem 2). These properties do not hold for general GP approximations (as illustrated for SVGP in Figure 1 and discussed below).
>
> > The second [missing discussion] is distinguishing the combined uncertainty (in eqn. 6) from the regular approximate posterior variance at test points.
>
> **The combined covariance is a special case of the approximate posterior covariance, _but_ the specific properties of $C_i$ are crucial.**
>
> The approximate posterior std. deviation generally *cannot* be decomposed into a term bounding the error to the latent function (combined uncertainty) and a term bounding the error to the mathematical posterior mean (computational uncertainty). This is only possible if $L \approx \hat{K}^{-1}$ satisfies eqn. (S42) (used in the proof of Theorem 2 in lines 704 and 706), as is the case for IterGP where $L=C_i$.
>
>
> ### Q3: Applicability of Theorem 2 to Other GP Approximations
>
> > [...] since other approximate GP inference techniques have their own approximation of $\hat{K}^{-1}$, their approximate posterior variance can also be used to give worst-case bounds in the sense of Theorem 2.
>
> **Theorem 2 does not apply to other approximate GP techniques.**
>
> Most GP approximations (Nyström / SVGP, RFF, NNGP) do *not* satisfy eqn (S42) and therefore *do not satisfy Theorem 2*. If Theorem 2 were to apply to SVGP, then SVGP would *provably* not underestimate uncertainty. This would directly contradict the literature on SVGP [Bauer2016, Huggins2019] and the illustration in Figure 1. However, there may be a close IterGP analog, which improves the UQ as we show for SVGP -> IterGP-PI.
>
> > [...] Skimming through Theorem 2’s proof, there is no part of the proof that relies on the actual structure of IterGP, in the sense that the statement will hold if we replace $C_i$ with any other approximation of $\hat{K}^{-1}$.
>
> **This is incorrect. The proof of Theorem 2 relies on the fact that $C_i \hat{K} C_i = C_i$ (eqn S42).** See lines 704 and 706.
>
> Eqn. (S42) is satisfied if $C_i\hat{K}$ is the $\hat{K}$-orthogonal projection onto the space spanned by the actions. Algorithm 1 precisely constructs this projection for a given sequence of actions. Since orthogonal projections are unique, *if another GP approximation $L \approx \hat{K}^{-1}$ is such a projection, it is an instance of IterGP.*
>
>
> ### Other
> > Lines 83-84: Equation 2 is true if the prior mean is zero.
>
> Thanks for pointing this out! We've added the missing prior mean in the final version.
>
> > Lines 96:97: the contraction only happens for a well-chosen set of actions $s_i$, not in general
>
> The posterior contracts for any policy, which generates linearly independent actions $s_i$ (see Proposition S4).
>
> > Lines 119-120: it’s not true that the algorithm is online. Line 9 of Algorithm 1 requires the whole kernel matrix $\hat{K}$
>
> **Algorithm 1 is online as we prove in Theorem S7**, but can be more concisely written via the whole kernel matrix.

---

> > ### Comment · Reviewer_pTot · 2022-08-03
> > **Discussion**
> >
> > I thank the authors for addressing my concerns. I am happy with the explanation that Theorem 2 is applicable to kernel approximations that satisfy S42. I would like to see the authors update the main text to include such subtleties. I have updated my score.

---

> > > ### Author Response · Authors · 2022-08-03
> > > **Rebuttal Response**
> > >
> > > Thank you for your prompt response. We will make sure to include your suggested discussion points in the final version of the paper and will elaborate on their subleties.

---

### Official Review · Reviewer_Cb8D · 2022-07-08

**Rating:** 6
**Confidence:** 3
**Soundness:** 2 fair
**Presentation:** 2 fair
**Contribution:** 3 good

**Summary:**

        This paper proposed a method to take into account extra uncertainty arising from using approximate GP solutions to scale GPs to large datasets. The proposed method is based on making inference about some parts of the computation of the posterior predictive distribution of a full GP. The authors indicate that the proposed approach is a generalization of many GP approximate methods. Theoretical results associated to the proposed method and derived and some experiments are carried out on synthetic and real-world data.


**Questions:**

It is not clear that the marginalization of v* below Eq. (2) gives the GP prior. The mean would be zero mean after the marginalization, using Eq. (2). The resulting variance will have to add the variance of the mean. Can the authors clarify this? It seems Eq. (2) is missing the GP mean.

Above Eq. (3) should not r_i bet r_i-1?


**Limitations:**

The authors have indicated that they have described the limitations of their method. However, they do not indicate where in the manuscript.


**Strengths And Weaknesses:**

Weaknesses:

        - The explanation of the proposed method is poor. The authors have to make a better job at explaining their method. Eq. (3) and the update on p(v*) has to be better explained.

        - Section 2.1 is poorly explained. It is not clear the connections with other methods.

        - The experimental section is a bit weak.

Strengths:

        - Nice theoretical results are derived for the proposed method.

---

> ### Author Response · Authors · 2022-08-02
> **Author Response to Reviewer Cb8D**
>
>
> Thank you for your review and the suggestions for improvement. We will use the extra page available for the final version to expand the explanation of how our method is derived and the connection to other GP approximations. We hope to answer any raised questions below. If anything should remain unclear, we would be happy to follow-up during the discussion period.
>
> ## Detailed Response
>
> ### Questions
> > - It is not clear that the marginalization of $v_*$ below Eq. (2) gives the GP prior. The mean would be zero mean after the marginalization, using Eq. (2). The resulting variance will have to add the variance of the mean. Can the authors clarify this? It seems Eq. (2) is missing the GP mean.
>
> Thank you for pointing out the typo in eqn. (2). It should contain the prior mean function $\mu(x_\star)$ and therefore read
>
> $$p(f_\star \mid v_*) = \mathcal{N}(\mu(x_\star) + k(x_\star, X)v_*, k_*(x_\star, x_\star))$$
>
> Now, if we marginalize out $p(v_*) = \mathcal{N}(v_*; 0, \hat{K}^{-1})$, then the resulting marginal $p(f_\star) = \int p(f_\star \mid v_*)p(v_*)dv_*$ has mean $\mu(x_\star) \eqqcolon \mu_\star$ and covariance
>
> $$k_*(x_\star, x_\star) + \underbrace{k(x_\star, X) \hat{K}^{-1} k(X, x_\star)}_{\text{uncertainty from marginalization}} = k(x_\star, x_\star) - k(x_\star, X) \hat{K}^{-1} k(X, x_\star) + k(x_\star, X) \hat{K}^{-1} k(X, x_\star) = k(x_\star, x_\star) \eqqcolon K_\star$$
>
> The mean and covariance after marginalization are now precisely the prior $\mathcal{GP}(\mu, k)$ evaluated at the new datapoints $x_\star$. Therefore it should correctly read in l87: "recovers the GP prior $\mathcal{N}(\mu_\star, K_\star)$."
>
> We realize that the visual differentiation between $\*$ and $\star$ may be difficult. To avoid confusion, we will update our notation to more clearly distinguish between the mathematical posterior (denoted by $*$) and new data points to predict on (denoted by $\star$).
>
> > - Above Eq. (3) should not $r_i$ be $r_{i-1}$?
>
> Yes it should be. Thank you for pointing this out.
>
> ### Other
> > The authors have indicated that they have described the limitations of their method. However, they do not indicate where in the manuscript.
>
> We describe limitations of IterGP as compared to linear time approximations in Section 2.2: "The Cost of Computational Uncertainty".

---

> > ### Comment · Reviewer_Cb8D · 2022-08-08
> > **Response to Authors**
> >
> > Thank you for the detailed explanations. I will keep my score as it is.

---

### Official Review · Reviewer_r5qg · 2022-07-12

**Rating:** 7
**Confidence:** 5
**Ethics Flag:** Yes
**Soundness:** 3 good
**Presentation:** 3 good
**Contribution:** 3 good

**Summary:**

Gaussian process regression model is impractical for big data and various approximations have been proposed to alleviate such difficulty. However, people have overlooked the uncertainty due to the numerical approximations for saving computational resources. This paper presented a novel low-complexity algorithm that respects uncertainties arising from the finite number of data observed and the finite amount of computation expended. Besides, theoretical analyses are given to support the effectiveness of the proposed algorithm. Experimental evaluations were also conducted to verify the proposed algorithm with some widely used approximation methods, such as sparse GP.

**Questions:**

1. Would the proposed algorithm work for distributed GP as well? If yes, what is the action $s$_i?
2. The theoretical analysis assumes that the actions $s$_i are linearly independent. Is this always true for the considered approximations?
3. Have you verified that the "computational uncertainty" term will go to zero when the computational resource is infinite?
4. Do you have more experimental results obtained using other kernels, such as the spectral mixture kernel and some hybrid kernels?
5. In the conclusion, you mentioned that the proposed algorithm is particularly useful for online data processing, which is not clear to me?


**Ethics Review Area:**

["I don’t know"]

**Limitations:**

Maybe more experiments with different kernel functions and different data lengths would be helpful.

**Strengths And Weaknesses:**

Strengths:
1. A novel GP approximation that accounts for computational uncertainty.
2. Well-crafted performance analyses (Th.1 and Th.2)
3. Experimental evaluations in terms of a few widely used GP approximations.

Weaknesses:
1. In Eq.(6), I understand that the "computational uncertainty" term results in a combined uncertainty that is cheaper to evaluate; however, I doubt this term is solely due to the numerical approximation made to the standard GP.

---

> ### Author Response · Authors · 2022-08-02
> **Author Response to Reviewer r5qg**
>
> Thank you for your positive review. We are pleased you recognize the strengths of our paper. We attempt to answer any remaining questions below. If anything remains unclear, we would be happy to clarify during the discussion period.
>
> ## Detailed Response
>
> ### Questions
>
> > 1. Would the proposed algorithm work for distributed GP as well? If yes, what is the action $s_i$?
>
> Assuming distributed compute nodes are available, a direct way to leverage them for IterGP is to perform distributed matrix-vector multiplication. It's an interesting question to consider whether one can recover distributed Gaussian processes [Deisenroth2015] in the IterGP framework. In [Deisenroth2015], each compute node has access to part of the dataset, individual GP experts are trained and then combined. Training on only a subset of the dataset in the IterGP framework corresponds to actions $s_i$ which have non-zero entries only for the entries corresponding to those datapoints in the subset. We conjecture that similar to the distributed framework in [Deisenroth2015], the combination of the information propagated up from the individual nodes could prove challenging.
>
> > 2. The theoretical analysis assumes that the actions $s_i$ are linearly independent. Is this always true for the considered approximations?
>
> This is true by construction for IterGP-Chol, IterGP-PBR and IterGP-CG. For IterGP-PI's actions $s_i = k(X, z_i)$, the vectors may be dependent for adversarially chosen inducing points (i.e. $z_1 = z_2$).
>
> > 3. Have you verified that the "computational uncertainty" term will go to zero when the computational resource is infinite?
>
> For $n$ linearly independent actions the computational uncertainty goes to zero in at most $n$ iterations. To see this recognize that $C_n = \hat{K}^{-1}$ and therefore the computational uncertainty $\Sigma_n = k(x_\star, X) \Sigma_n k(X, x_\star) =  k(x_\star, X) (\hat{K}^{-1} - C_n) k(X, x_\star) = 0$ in eqn (6).
>
> > 4. Do you have more experimental results obtained using other kernels, such as the spectral mixture kernel and some hybrid kernels?
>
> Any further experiments we performed beyond the main paper are in the appendix in Section S3. To be concise, we focused on the most commonly used kernels: RBF, Matern(1/2) and Matern(3/2).
>
> > 5. In the conclusion, you mentioned that the proposed algorithm is particularly useful for online data processing, which is not clear to me?
>
> Running IterGP on a large dataset with actions not targeting part of the dataset at all (i.e. having zero entries for the corresponding datapoints) is equivalent to not having observed that data yet. In that sense, Algorithm 1 is inherently online. We make this precise in Theorem S7.
>
> ### References
> - [Deisenroth2015] Deisenroth, Marc, and Jun Wei Ng. "Distributed Gaussian processes." International Conference on Machine Learning (ICML), 2015.

---

### Official Review · Reviewer_5Mot · 2022-07-14

**Rating:** 7
**Confidence:** 2
**Soundness:** 3 good
**Presentation:** 3 good
**Contribution:** 4 excellent

**Summary:**

The authors present a novel method, named IterGP, for doing approximate inference in Gaussian processes. Taking a probabilistic numerics approach, they treat the representer weights, $\mathbf v_* = \hat{\mathbf K}^{-1} (\mathbf y - \boldsymbol \mu)$, as a quantity to iteratively compute. By building on recent advances in probabilistic linear solvers, they derive an algorithm for iteratively updating a distribution over the representer weights, which captures the (computational) uncertainty in them.
By reparameterising the GP posterior in terms of the representer weights and then marginalising them out, the authors obtain a new GP posterior expression, which implicitly accounts for the uncertainty in approximation made by the iterative representer weights.

While explicitly computing the computational uncertainty is expensive, computing the combined (computational and mathematical) uncertainty has a quadratic cost, meaning that IterGP computationally sits in-between exact GP inference (cubic) and linear-time approximation methods, which, however, do not provide calibrated uncertainty estimates. At the same time, the memory cost of IterGP is only linear.

To make the iterative updates, IterGP requires a policy for selecting and weighing data. The authors discuss different strategies and show how they correspond to standard approximation methods, such as conjugate gradients and inducing point methods, and extend these to properly account for computational uncertainty.

The paper concludes with a theoretical analysis, proving general convergence of the posterior mean as well as proving that the combined uncertainty is a worst-case bound on the error, and an empirical evaluation of IterGP, showing that it provides better posterior estimates than commonly used methods.


**Questions:**

NB I have quite limited experience with probabilistic numerics, so the following questions should be read in this context! :)

1. I don't fully understand the definition of the residual in line 90. Specifically, why does it contain $\hat{\mathbf K}$ and not just $\mathbf K$? I would have expected the latter if it's the residual between the observations and the GP posterior mean.

2. I found the discussion of the overconfidence in SVGP extremely interesting, but I don't quite understand the point made in lines 142-146. Since the representer weights of the inducing points are typically calculated exactly, where does the error in the uncertainty estimate (i.e., the overconfidence) come from? (To be clear, this is absolutely not a crucial question; I'm just trying to understand SVGP :) ).

3. I suppose the choice of a stopping criterion will be problem-specific, but do you have some suggestions for this? While the computational uncertainty is too expensive to explicitly compute in general, is it possible to somehow estimate it and stop when we have reached a pre-defined tolerance?

4. Figure S5 (b) in the supplementary hints are instabilities in certain settings. Do you know which parts of the algorithm lead to this, and can you say something more about when this could happen?


**Limitations:**

Potential negative societal impacts have not been discussed, but it is not necessary given the paper's theoretical nature.

**Strengths And Weaknesses:**

**Strengths**

The paper overall is of very high quality. The writing is clear, and the figures are fantastic. I especially like the pedagogical use of colours for the different kinds of uncertainties, which are shared between figures, equations, and the text.

From a technical perspective, the paper is very thorough, and the authors carefully explain the intuition behind the method where possible. They also provide a good discussion of the different policies for choosing actions and how these link back to commonly used approximation methods, showing a kind of unification (and extension) of these methods.

The theoretical analysis seems solid, but I did not check it in detail.

While the field of probabilistic numerics is not my area of expertise, the work seems original and highly novel to me. Overconfident posteriors are a well-known issue in the approximate GP world, so the contributions in this paper make it a very exciting piece of work. It is definitely a significant contribution to the approximate GP literature, which will be of major interest to the NeurIPS community.

**Weaknesses**

While the writing is overall of very high quality, there are a lot of new concepts to digest for someone not in the field of probabilistic numerics (but who does have experience with GPs). A short background on probabilistic numerics (perhaps probabilistic linear solvers in particular) would have been helpful for me, but I understand that I might not be the target audience.

One thing that the authors did not discuss is how to choose the stopping criterion in algorithm 1. I suppose there are some trivial choices (e.g., if one has a fixed computational budget), but perhaps one can be smarter here. A brief discussion of this would have been nice.

For the experiments, it would have been interesting to see SVGP and IterGP compare in terms of wall-clock time for different numbers of inducing points/iterations. Since SVGP is faster to compute, I suppose it would achieve better results for very few inducing points. It would be interesting to see when IterGP begins to perform better for a fixed computational budget.

Also, it would be interesting to see the RMSE and NLL obtained by an exact GP indicated in both figure 4 and 5.

**Various minor things**
* In equation (2), I think a $+ \mu$ is missing in the mean.
* The next-to-last sentence in the caption of figure 2 appears incomplete.
* A couple of lines in algorithm 1 are coloured grey, but there's no discussion of what this means, as far as I can see.

---

> ### Author Response · Authors · 2022-08-02
> **Author Response to Reviewer 5Mot**
>
>
> Thank you for your positive feedback! As requested we will expand the background section in the final version. We hope we could answer any questions you raised below. If anything should remain unclear, we would be glad to address it in the discussion period.
>
> ## Detailed Response
>
> ### Questions
> >  1. I don't fully understand the definition of the residual in line 90. Specifically, why does it contain $\hat K$ and not just $K$. I would have expected the latter if it's the residual between the observations and the GP posterior mean.
>
> The term "residual" is unfortunately overloaded in this context. The residual $r_i = y - \hat{K} v_i$ as referred to in the paper is the *residual of the approximate solution $v_i$ to the linear system* $\hat{K} v_* = y$, which defines the representer weights. This residual goes to zero as IterGP runs, no matter the observational noise $\sigma^2$. In contrast the *residual of the GP prediction* $y - \mu_i(X) = y - Kv_i$ does not.
>
> > 2. I found the discussion of the overconfidence in SVGP extremely interesting, but I don't quite understand the point made in lines 142-146. Since the representer weights of the inducing points are typically calculated exactly, where does the error in the uncertainty estimate (i.e., the overconfidence) come from?
>
> In the paper we view the representer weights as unknown initially and, as we expend more computation, uncertainty about them contracts. The current estimate for the representer weights is given by $v_i = C_i (y-\mu) = S_i(S_i^\top \hat{K} S_i)^{-1}S_i^\top (y-\mu)$ (see eqn (4) and line 95). This is a Bayesian update on the representer weights where $S_i^\top \hat{K} S_i = S_i^\top \hat{K} \Sigma_0 \hat{K} S_i$ is the Gram matrix. Its interpretation is how surprising the projections of the representer weights $S_i^\top (y-\mu) = S_i^\top \hat{K}v_*$ should be to us, given our prior uncertainty $\Sigma_0$ about the representer weights. Choosing $S_i=k(X, Z)$ gives the form of IterGP-PI's posterior mean in eqn (8). Now, SVGP uses a similar form for the posterior mean $\mu(\cdot)$ (see eqn (7)), **but** the Gram matrix is "smaller", i.e. $q(X, X) \preceq k(X, X)$ and therefore $K_{ZX}(q(X, X) + \sigma^2I)K_{XZ} \preceq K_{ZX}(k(X, X) + \sigma^2I)K_{XZ} = S_i^\top \hat{K} S_i$. Therefore SVGP in its update of the representer weights is *not uncertain enough* as the Bayesian update would require, leading to its overconfidence in its posterior mean estimate.
>
> > 3. I suppose the choice of a stopping criterion will be problem-specific, but do you have some suggestions for this?
>
> In our implementation we use
> - *the absolute and relative norm of the residual*, i.e. $\lVert r_i \rVert_2 < \max(\delta_{\mathrm{reltol}} \lVert y \rVert_2, \delta_{\mathrm{abstol}})$, which terminates if a sufficient accuracy is reached on the training data, and
> - *a maximum number of iterations*, which defines the computational budget.
>
> As you point out one could specify others. For example, from a probabilistic numerics viewpoint one may want to stop if the combined marginal uncertainty at the training data is sufficiently small (relative to the observation noise), i.e. $\operatorname{tr}(K - K C_i K) = \operatorname{tr}(K) - \operatorname{tr}(\tilde{D}_i^\top K K \tilde{D}_i) < \delta_\mathrm{unctol}$. Note that the vectors in the matrix $K \tilde{D}_i \in \mathbb{R}^{n \times i}$ are computed anyway as part of Algorithm 1. Storing them to use as a stopping criterion does not affect the asymptotic memory cost (which is $\mathcal{O}(ni)$) and computing the stopping criterion has time complexity $\mathcal{O}(ni^2)$.
>
> > 4. Figure S5 (b) in the supplementary hints are instabilities in certain settings. Do you know which parts of the algorithm lead to this, and can you say something more about when this could happen?
>
> We've empirically only observed this so far for IterGP-PI, i.e. the choice of actions which most closely corresponds to SVGP on the KEGGundir dataset (see Figure S5b). We conjecture this may happen if the inducing points are chosen in such a way such that $S_i=K(X, Z)$ is numerically close to having rank $<i$. Algorithm 1 implicitly orthogonalizes the actions $S_i$ with respect to the $\hat K$-inner product. We conjecture that our current implementation may become numerically unstable if the assumption of independence of the actions is numerically close to not being satisfied. In this case the error introduced by finite precision potentially affects the result of the reorthogonalization in step 8 of Algorithm 1.
>
> ### Other
> > -  A couple of lines in algorithm 1 are coloured grey, but there's no discussion of what this means [...].
>
> Greyed out quantities are *not necessary to compute the GP approximation*. We included them since their computation might be of independent interest (the kernel matrix approximation $Q_i$), or because they correspond to quantities we introduce in the derivation of the Algorithm (the belief over the representer weights $p(v_*)$).

---

> > ### Comment · Reviewer_5Mot · 2022-08-04
> > **Thank you for the rebuttal**
> >
> > Thank you for addressing my comments and questions - things are much clearer now. In particular, thank you for elaborating on the SVGP overconfidence.
> >
> > I suggest you include the details on your stopping criterion in a revised paper (adding it to the supplementary should be just fine). It's both interesting and relevant to future users of IterGP.
> >
> > I also think it would improve your paper if you added the short clarification of the grey lines in algorithm 1 to the main paper. It is great that you included the lines and marked them, but it was just a bit confusing to me.
> >
> > I still think this is excellent work and will keep my score.

---

> > > ### Author Response · Authors · 2022-08-04
> > > **Thank you**
> > >
> > > Thank you for your response and feedback! We will make sure to include the details on the stopping criteria used and a clarification about the grey lines in Algorithm 1 in the final version.

---

### Meta-Review · Area_Chair_zQPE · 2022-08-22

**Recommendation:** Accept
**Confidence:** Certain

**Metareview:**

Gaussian Processes are a very nice modelisation tool in Bayesian nonparametrics, with very nice uncertainty quantification. But they also lead to serious computational issues. So, in practice, it is difficult to know what part of the uncertainty is due to the data, and what is due to approximations in the computations. Here, the authors propose a new (and cheap) iterative approximation, IterGP. They analyse carefully its approximation error in Section 3. Experimental results corroborate this analysis. Overall, IterGP can reach the same accuracy than previous methods with a limited number of steps, and thus a smaller computational burden. Increasing the number of steps will of course make it even more accurate.

The four reviewers agreed on the relevance of the algorithm, the quality of its technical analysis and the quality of the experimental results (I agree with them). Reviewer 5Mot praised the high quality of the writing. The other reviewers overall agreed, but provided a list of minor points that could be fixed to improve the paper. I therefore recommend to accept this paper.

**Award:**

No

---

### Decision · Program_Chairs · 2022-09-14

Accept